# Graph-based Nearest Neighbor Search in Hyperbolic Spaces

**Liudmila Prokhorenkova**
Yandex Research
Moscow, Russia
ostroumova-la@yandex.ru

**Dmitry Baranchuk**
Yandex Research
Moscow, Russia
dbaranchuk@yandex-team.ru

**Nikolay Bogachev**
IITP RAS; MIPT
Moscow, Russia
nvbogach@mail.ru

**Yury Demidovich**
MIPT
Moscow, Russia
demidovich.yua@phystech.edu

**Alexander Kolpakov**
Université de Neuchâtel
Neuchâtel, Switzerland
kolpakov.alexander@gmail.com

## Abstract

The nearest neighbor search (NNS) problem is widely studied in Euclidean space, and graph-based algorithms are known to outperform other approaches for this task. However, hyperbolic geometry often allows for better data representation in various domains, including graphs, words, and images. In this paper, we show that graph-based approaches are also well suited for hyperbolic geometry. From a theoretical perspective, we rigorously analyze the time and space complexity of graph-based NNS, assuming that an $n$-element dataset is uniformly distributed within a $d$-dimensional ball of radius $R$ in the hyperbolic space of curvature $-1$. Under some conditions on $R$ and $d$, we derive the time and space complexity of graph-based NNS and compare the obtained results with known guarantees for the Euclidean case. Interestingly, in the dense setting ($d \ll \log n$) and under some assumptions on the radius $R$, graph-based NNS has *lower* time complexity in the hyperbolic space. This agrees with our experiments: we consider datasets embedded in hyperbolic and Euclidean spaces and show that graph-based NNS can be more efficient in the hyperbolic space. We also demonstrate that graph-based methods outperform other existing baselines for hyperbolic NNS. Overall, our theoretical and empirical analysis suggests that graph-based NNS can be considered a default approach for similarity search in hyperbolic spaces.

## 1 Introduction

Given a dataset $\mathcal{D}$ of $n$ elements and an arbitrary query $\mathbf{q}$, the goal of the nearest neighbor search (NNS) is to quickly retrieve one or several elements from $\mathcal{D}$ closest to $\mathbf{q}$. NNS is extensively used in a wide range of machine learning systems: for non-parametric classification and regression, language modeling, information retrieval, recommendations, and many others (Bishop, 2006; Chen et al., 2018; May & Ozerov, 2015; Shakhnarovich et al., 2006). In large-scale applications, searching for the nearest neighbors can be a bottleneck, so this operation's efficiency is of the essence.

Many efficient NNS algorithms exist for Euclidean and, generally, $\ell_p$ spaces. Among them are methods based on recursive partitions of the space, e.g., k-d trees and random projection trees (Bentley, 1975; Dasgupta & Freund, 2008; Dasgupta & Sinha, 2015; Keivani & Sinha, 2018), well studied Locality Sensitive Hashing (LSH) (Indyk & Motwani, 1998), and methods based on similarity graphs (Fu et al., 2019; Iwasaki & Miyazaki, 2018; Malkov et al., 2014; Malkov & Yashunin, 2018).

However, hyperbolic geometry is often found to be more suitable for data representation in various domains, including natural language processing, computer vision, graph analysis, and recommendations (Chamberlain et al., 2017; Chami et al., 2020; Khrulkov et al., 2020; Mirvakhabova et al., 2020; Nickel & Kiela, 2017; 2018; Tifrea et al., 2019). Intuitively, the hyperbolic space has advantages over standard representations when data has a latent hierarchical structure which is often the case. Hence, a natural question is: how to perform efficient NNS in hyperbolic spaces? This problem has started to gain attention recently and was addressed in Wu & Charikar (2020). In a nutshell, Wu & Charikar (2020) show how any given *Euclidean* NNS algorithm can be applied to the *hyperbolic* space. To get convergence guarantees, one has to call the chosen NNS method several times, which leads to increased NNS complexity compared to the original Euclidean algorithm. In the current paper, we choose an alternative approach and apply a leading NNS technique *directly* to the hyperbolic space. We demonstrate that under some assumptions, the proposed method does not have increased complexity and may even *reduce* the query time compared to its Euclidean counterpart.

We focus on graph-based NNS methods that are based on a best-first routing over similarity graphs. Namely, at the pre-processing phase, a similarity graph is constructed on the elements of the dataset. Then, given a query $q$, we start from some element of $\mathcal{D}$ and measure distances between its neighbors and $q$. The neighbors are added to a priority queue. Then, we pick the closest to $q$ candidate from this queue and continue the process (Fu et al., 2019; Iwasaki & Miyazaki, 2018; Malkov et al., 2014; Malkov & Yashunin, 2018). It was found empirically that graph-based NNS outperforms other approaches in many large-scale applications (Aumüller et al., 2019). In contrast to other NNS methods, graph-based approaches can be easily applied to non-Euclidean spaces. Indeed, for most algorithms, both graph construction and graph routing can be performed without appealing to the geometry of the underlying space — only the relative ordering of neighbors is used. Thus, one can replace Euclidean distance with the hyperbolic one in a favorite graph-based approach. However, it is a priori not known how graph-based approaches would work in such a non-standard scenario.

In this paper, we show that graph-based approaches are well suited for NNS in hyperbolic spaces. From a theoretical perspective, we analyze the time and space complexity of graph-based NNS, assuming that the data elements are distributed uniformly within a ball of radius $R$ in the hyperbolic space of curvature $-1$. In this regard, we compare the obtained results with Prokhorenkova & Shekhovtsov (2020) who studied graph-based algorithms for uniform distribution of elements over a $d$-dimensional sphere. In the dense setting (assuming that $d \ll \log n$ and $R$ does not grow too fast), we derive the time and space complexity using the fact that locally hyperbolic graph-based NNS behaves similarly to the Euclidean case. However, globally, graph-based search in hyperbolic spaces requires *fewer search steps*. Thus, hyperbolic NNS can be even *more efficient* compared to its Euclidean counterpart. In the sparse setting ($d \gg \log n$), we show that hyperbolic NNS has similar complexity to the Euclidean case.

From a practical perspective, we first illustrate our theoretical findings via synthetic experiments on uniform data. Then, we analyze graph-based hyperbolic NNS in practical applications where our theoretical results cannot be directly applied. For this, we consider Poincaré GloVe word embeddings (Tifrea et al., 2019) that are known to outperform the standard representation for tasks of similarity, analogy, and hypernymy detection. We compare graph-based NNS with existing baselines (Wu & Charikar, 2020) and show that graph-based approaches have significantly better performance. We also compare the performance of graph-based NNS for Euclidean and hyperbolic representations of the same sets of elements and observe that hyperbolic NNS can be more efficient.

## 2 Preliminaries

**Graph-based nearest neighbor search** Graph-based algorithms for NNS are known to outperform other approaches in many large-scale applications (Aumüller et al., 2019). The main idea is to construct a proximity graph at the pre-processing stage and traverse this graph towards the query $q$ at query time. The best-first search is usually used for traversal: we maintain a priority queue, and at each step, we pick the closest to $q$ candidate from this queue, measure distances between its neighbors and $q$, and add the neighbors to the queue. Many algorithms are based on this idea (Baranchuk et al., 2019; Fu et al., 2019; Iwasaki & Miyazaki, 2018; Malkov & Yashunin, 2018).

Unfortunately, despite their success in practice, graph-based approaches do not have solid theoretical guarantees. This problem was first approached by Laarhoven (2018) who provided time–space

trade-offs for approximate nearest neighbor (ANN) search in sparse datasets ($d \gg \log n$) uniformly distributed on a $d$-dimensional Euclidean sphere. This work considered nearest neighbor graphs, where each point is connected to its neighborhood lying within a ball of a fixed radius, and assumed greedy graph traversal. Prokhorenkova & Shekhovtsov (2020) extended this result: for the uniform distribution, the authors consider both sparse ($d \gg \log n$) and dense ($d \ll \log n$) regimes, and they analyze the effect of two improvements commonly used in practice: replacing greedy search by best-first (or beam) search and adding long edges to speed up graph traversal at early stages.

In this paper, we aim at understanding graph-based NNS in hyperbolic spaces. For this, we adopt a similar setup to Prokhorenkova & Shekhovtsov (2020), but instead of a uniform distribution over a sphere, we assume a uniform distribution within a ball of a certain radius in the hyperbolic space. The obtained time and space complexity heavily relies on this radius: for small values, the obtained results are similar to the spherical case, while for larger values, the exponential growth of hyperbolic volumes allows us to obtain even faster convergence in the dense regime.

**Hyperbolic geometry**  Now, let us discuss the basics of hyperbolic geometry. Let $\mathbb{R}^{d,1}$ be the $(d + 1)$-dimensional *Minkowski space*, i.e., the real vector space $\mathbb{R}^{d+1}$ equipped with the inner product of signature $(d, 1)$:

$$\langle \mathbf{x}, \mathbf{y} \rangle_h = -x_0 y_0 + x_1 y_1 + \ldots + x_d y_d. \tag{1}$$

The *vector model* of the *$d$-dimensional hyperbolic Lobachevsky space* $\mathbb{H}^d$ is the connected component of the standard two-sheeted hyperboloid contained in the *future light cone*: $\mathbb{H}^d = \{ \mathbf{x} \in \mathbb{R}^{d,1} \mid \langle \mathbf{x}, \mathbf{x} \rangle_h = -1, x_0 > 0 \}$. The hyperbolic metric is given by $\cosh \rho(\mathbf{x}, \mathbf{y}) = -\langle \mathbf{x}, \mathbf{y} \rangle_h$. From the definition of the hyperbolic space, we see that $x_0 = \sqrt{1 + x_1^2 + \ldots + x_d^2}$. The volume element in $\mathbb{H}^d$ is $\frac{dx_1 \ldots dx_d}{\sqrt{1 + x_1^2 + \ldots + x_d^2}}$ (Ratcliffe, 2019).

Hyperbolic space is known to be useful for data representation in various domains. It is especially convenient for embedding hierarchical data (Nickel & Kiela, 2017). The reason is that the volume of a ball in the hyperbolic space grows exponentially with its radius, in contrast to the Euclidean space, where the growth is polynomial. Formally, let $B(\mathbf{x}, r)$ be a closed hyperbolic ball centered at a point $\mathbf{x} \in \mathbb{H}^d$ of radius $r > 0$. Then, its volume is

$$\mathrm{Vol}_{\mathbb{H}^d}(B(\mathbf{x}, r)) = \mathrm{Vol}(\mathbb{S}^{d-1}) \int_0^r \sinh^{d-1}(t) \, dt \,, \tag{2}$$

where $\mathbb{S}^{d-1}$ is the unit $(d-1)$-sphere and $\mathrm{Vol}(\mathcal{M})$ denotes the full volume of a Riemannian manifold $\mathcal{M}$. That is, $\mathrm{Vol}(\mathbb{S}^{d-1}) = 2\pi^{d/2}/\Gamma(d/2)$. For $r \gg \log d$, we have

$$\mathrm{Vol}_{\mathbb{H}^d}(B(\mathbf{x}, r)) \sim \mathrm{Vol}(\mathbb{S}^{d-1}) \, e^{r(d-1)} \, \Theta(d^{-1} 2^{-d}). \tag{3}$$

The main term here is $e^{r(d-1)}$, so the ball volume $\mathrm{Vol}_{\mathbb{H}^d}(B(\mathbf{x}, r))$ grows exponentially when $r$ is large. However, for small hyperbolic balls (as $r \to 0$), we have $\sinh(r) \sim r$. Hence, the volume of a ball (2) behaves similarly to the Euclidean case. In Section 4.1, we provide a formal general result that shows that not only the above argument is correct, but also the volume of a measurable subset of a hyperbolic ball behaves asymptotically exactly like its Euclidean analog.

**Nearest neighbor search in hyperbolic space**  While there are many NNS algorithms for $\ell_p$ spaces, little is known about efficient hyperbolic analogs. Krauthgamer & Lee (2006) proved the existence of an approximate near neighbor algorithm for general geodesic spaces of hyperbolicity $\delta$. From a theoretical perspective, the bound is given in terms of the doubling dimension. For uniform datasets, we may conclude that the obtained time complexity is $2^{O(d)} \log^2 n$, while the constant in the exponent is not given. In contrast, we specify this term as we focus on $d \to \infty$. Another difference is that the algorithm of Krauthgamer & Lee (2006) requires $O(n^2)$ storage that is usually infeasible in practice. In contrast, graph-based algorithms require $n M^d$ for some constant $M$ if $d \ll \log n$. Finally, while the theoretical analysis is given in a very general setup, no implementation is provided and extracting an efficient implementation from the proof is challenging.

In a recent paper, Wu & Charikar (2020) reduce the problem of hyperbolic NNS to Euclidean NNS. Namely, assume that we have an efficient Euclidean algorithm and a dataset is represented in the

Poincaré ball model. Then, we can apply the algorithm to the Euclidean representation, but this may not give the correct nearest neighbor. Wu & Charikar (2020) propose several solutions allowing for obtaining either exact or approximate nearest neighbors. In all of them, the Euclidean NNS algorithm should be applied several times. One approach is to repeatedly apply the algorithm to the whole dataset, but after each run, appropriately change the query point to get better hyperbolic neighbors at subsequent iterations. The time complexity of this algorithm is $k + 1$ times larger than the Euclidean search if the Euclidean nearest neighbor for the query is the $k$-th nearest hyperbolic neighbor. Another approach is to splits the ball into thin annuli and performs the approximate Euclidean NNS for each of them separately. From a theoretical perspective, it is hard to compare our results since Wu & Charikar (2020) do not consider any particular algorithm. We cannot apply this general result to graph-based algorithms since existing guarantees for the Euclidean space assume a uniform distribution. However, if we consider the uniform distribution in the Poincaré model of the hyperbolic space, the corresponding Euclidean distribution is far from uniform, i.e., the guarantees cannot be transferred. From a practical perspective, we compare graph-based algorithms with the method of Wu & Charikar (2020) and show that graph-based NNS is more efficient.

## 3   SETUP AND NOTATION

The current paper aims at analyzing how hyperbolic geometry of the underlying space affects the time and space complexity of NNS. In this regard, we compare our results with Prokhorenkova & Shekhovtsov (2020), so we use a similar setup and notation for convenience.

To make the analysis feasible, Laarhoven (2018) and Prokhorenkova & Shekhovtsov (2020) assume that the dataset is uniformly distributed on a sphere. The uniformity assumption is fairly strong. In Prokhorenkova & Shekhovtsov (2020), this setup was partially motivated by techniques allowing to make a dataset more uniform while approximately preserving the distances (Sablayrolles et al., 2018), which are helpful in some practical applications. We additionally note that in hyperbolic spaces, the uniform distribution of nodes within a ball leads to structures that are similar to real ones: if we consider a geometric graph on such nodes, we obtain the power-law degree distribution, small diameter, and high clustering coefficient (Krioukov et al., 2010), which further supports the meaningfulness of the uniform distribution in our scenario. The compactness of the sphere simplifies the analysis in previous work as it allows for avoiding boundary effects. However, we cannot make this assumption in our setup, as a sphere in the hyperbolic space is isometric to a sphere in the Euclidean space. Hence, to use the properties of hyperbolic geometry, we assume that the dataset is uniformly distributed within a ball of some radius in the hyperbolic space.

Formally, consider a ball $B(R) \subset \mathbb{H}^d$ of radius $R > 0$. Assume that the dataset is a finite collection of points $\mathcal{D} = \{\mathbf{x}_1, \ldots, \mathbf{x}_n\} \subset \mathbb{H}^d$, where $\mathbf{x}_i \in \mathcal{D}$ are i.i.d. random vectors uniformly distributed in $B(R)$. For a given query $\mathbf{q} \in B(R)$, let $\bar{\mathbf{x}} \in \mathcal{D}$ be its nearest neighbor. Then, the goal of the exact NNS is to find $\bar{\mathbf{x}}$. For completeness, following Laarhoven (2018); Prokhorenkova & Shekhovtsov (2020), we also consider $c, r$-approximate near neighbor problem ($c, r$-ANN). In this scenario, it is assumed that $\mathbf{q}$ is *planted* uniformly within a fixed distance $r > 0$ from its nearest neighbor $\bar{\mathbf{x}}$ and the goal is to return any $\mathbf{x}'$ such that $\rho(\mathbf{q}, \mathbf{x}') \leq cr$ for a fixed value $c > 1$ (Chen et al., 2018). Formally, for the exact nearest neighbor search, we also assume that $\mathbf{q}$ is planted uniformly within a distance $r$ from some element $\mathbf{x}$. However, we do not restrict the radius $r$ (the only restriction is that $\mathbf{x}$ should be the nearest neighbor for $\mathbf{q}$).

We follow the standard assumption that the dimension $d = d(n)$ is an increasing function of $n$ (Chen et al., 2018). We mostly focus on the so-called dense regime, where $d \ll \log n$. For the sparse regime ($d \gg \log n$), we formally show that graph-based NNS in the hyperbolic space has similar complexity to NNS on the sphere.

Graph-based approaches usually imply constructing a nearest neighbor graph (or its approximation), where each element is connected to its $k$ nearest neighbors. As we deal with uniformly distributed datasets, we assume that each node $\mathbf{x}$ is connected to all nodes $\mathbf{y}$ at a distance $\rho(\mathbf{x}, \mathbf{y}) \leq \rho^*$ for some $\rho^*$ chosen at the preprocessing stage. This is more convenient for the analysis, while similar results can be obtained for $k$-NN graphs, see Appendix D. Hence, at the preprocessing stage, we construct a similarity graph using $\rho^*$. Next, we receive a query $\mathbf{q}$, sample a random node $\mathbf{x} \in \mathcal{D}$, and at each step of the graph-based greedy descent, we measure the distances between the neighbors of the current node and $\mathbf{q}$ and move to the closest neighbor while we can make such greedy steps.

# 4 CONVERGENCE AND COMPLEXITY OF GRAPH-BASED NNS

## 4.1 LOCAL PROPERTIES OF HYPERBOLIC SPACE

Further in this section, we give formal statements regarding the convergence and complexity of greedy graph-based algorithms. Convergence depends on the probability of making a step towards the query from a given element. This probability, in turn, depends on the volumes of some balls' intersections. As our proof shows, only 'local' properties are important for convergence for the dense regime. Let us show that under some conditions, the local properties in the hyperbolic space are similar to those in the Euclidean space.

Formally, let $\mathbf{z} = (1, 0, \ldots, 0)$ and let $T_{\mathbf{z}}\mathbb{H}^d = \{\mathbf{x} \in \mathbb{R}^{d,1} \mid x_0 = 1\} \simeq \mathbb{R}^d$ be the tangent space. Let $\hat{c}$ be the central projection $\hat{c} : \mathbb{H}^d \to T_{\mathbf{z}}\mathbb{H}^d \simeq \mathbb{R}^d$ from the origin $\mathbf{o} = (0, \ldots, 0)$. The following lemma holds.

**Lemma 1.** *Suppose that $U \subset \mathbb{H}^d$ is a measurable set such that $U \subset B(\mathbf{z}, r)$. If $r \ll \frac{1}{\sqrt{d}}$ and $d \to \infty$, then we have $\mathrm{Vol}_{\mathbb{H}^d}(U) = \mathrm{Vol}_{\mathbb{E}^d}(\hat{c}(U)) \left(1 - O\left(d\,r^2\right)\right)$.*

This result allows us to apply the bounds on volumes of Euclidean balls intersections to such volumes in the hyperbolic space for small radii.

## 4.2 INTUITION BEHIND THE CHOICE OF PARAMETERS

Our theoretical analysis is asymptotic, meaning that the number of elements $n$ tends to infinity. Hence, we assume that the dimension $d = d(n)$ and the ball radius $R = R(n)$ depend on $n$. This section discusses our restrictions on $d(n)$ and $R(n)$ and how they affect the analysis. Here, we do not aim at giving mathematically rigorous statements but rather explain underlying intuitions. The formal results can be found in Section 4.3.

Assuming that the dataset is uniformly distributed on a unit sphere, there are two principally different regimes: dense with $d \ll \log n$ and sparse with $d \gg \log n$ (Prokhorenkova & Shekhovtsov, 2020). In the dense regime on a sphere, the distance from any element to its nearest neighbor tends to zero as $n$ grows. In this case, the greedy algorithm reaches the nearest neighbor with probability $1 - o(1)$ if graph degree is $M^d$ for $M > \sqrt{2}$. In the sparse regime, informally speaking, all nodes are at a spherical distance about $\pi/2$ from each other, making the NNS much more challenging. In this case, the graph degree has to grow as $n^{\varphi}$ for some $\varphi > 0$ in order to solve the approximate near neighbor problem (Prokhorenkova & Shekhovtsov, 2020).

In this paper, we mainly focus on the dense regime with $d \ll \log n$. As discussed in (Prokhorenkova & Shekhovtsov, 2020), this setup is more realistic for uniform data. Indeed, while in practice datasets are often embedded into spaces of large dimension, they usually have much smaller *intrinsic* dimension $d' \ll d$ (Beygelzimer et al., 2006; Lin & Zhao, 2019). It is also important to note that hyperbolic spaces are known to be beneficial when the dimension is small (Gu et al., 2019; Shevkunov & Prokhorenkova, 2021). For completeness of the study, we also analyze the sparse regime and theoretically show that under some conditions on $R$ graph-based NNS in the hyperbolic space has similar time and space complexity to NNS on the sphere.

We assume that the dataset is uniformly distributed within a ball $\mathcal{B} = B(R)$ in the hyperbolic space. Changing the radius $R$ is equivalent to changing the curvature of the space (Petersen, 2016). So, we may either assume that the curvature $C = -1$ is fixed while the radius of the ball $R = R(n)$ may change or assume that the radius is fixed while the curvature $C = C(n)$ changes. We further assume that $C = -1$ is fixed. Thus, we have three parameters: the number of elements $n$, the dimension $d = d(n)$, and the radius $R = R(n)$.

Denote by $V_d(R)$ the volume of $\mathcal{B}$, i.e., $V_d(R) := \mathrm{Vol}_{\mathbb{H}^d}(B(R))$ and by $v_d(R)$ the normalized volume, i.e., $v_d(R) := V_d(R)/\mathrm{Vol}(\mathbb{S}^{d-1})$. Let us analyze (approximately) the distance $\alpha$ from an element to its nearest neighbor. Intuitively, we should have

$$\frac{\mathrm{Vol}_{\mathbb{H}^d}(B(\alpha))}{\mathrm{Vol}_{\mathbb{H}^d}(B(R))} \approx \frac{1}{n}, \quad \mathrm{Vol}_{\mathbb{H}^d}(B(\alpha)) \approx \frac{V_d(R)}{n}. \tag{4}$$

Assume for now that $\alpha \ll 1/\sqrt{d}$ (approximately Euclidean setup, see Section 4.1). In this case,

$$\text{Vol}_{\mathbb{H}^d}(B(\alpha)) \sim \text{Vol}_{\mathbb{E}^d}(B(\alpha)) = \frac{\text{Vol}(\mathbb{S}^{d-1})\alpha^d}{d}.$$

Therefore, from (4),

$$\alpha^d \approx \frac{V_d(R)\,d}{n\,\text{Vol}(\mathbb{S}^{d-1})} = \frac{v_d(R)\,d}{n}.$$

Let us find an upper bound on the radius $R$ such that the condition $\alpha \ll 1/\sqrt{d}$ is satisfied. Note that $v_d(R) = O\left(e^{R(d-1)} \cdot 2^{-d} \cdot d^{-1}\right)$ (see Appendix B). Thus, we need

$$\left(\frac{v_d(R)\,d}{n}\right)^{1/d} \ll \frac{1}{\sqrt{d}}, \quad \left(\frac{e^{R(d-1)}\,2^{-d}}{n}\right)^{1/d} \ll \frac{1}{\sqrt{d}}, \quad e^R \ll n^{1/(d-1)} \cdot d^{-1/2}.$$

Based on that, we further assume that in the dense regime, $e^R \ll n^{1/(d-1)} \cdot d^{-1/2}$. In this case, the distance to the nearest neighbor is $\alpha = o\left(1/\sqrt{d}\right)$, and locally the graph-based search behaves similarly to the Euclidean case. We empirically show that our assumption on $R$ is reasonable: larger values of $R$ lead to less efficient NNS.

## 4.3 MAIN RESULTS

In this section, we formally analyze the performance of graph-based NNS for uniform datasets in the hyperbolic space. As discussed above, to construct the nearest neighbor graph, we connect each point to its neighborhood located within a ball of a fixed radius. In the dense regime, for any constant $M > 1$, let $G(M)$ be a graph obtained by connecting $\mathbf{x}_i$ and $\mathbf{x}_j$ iff

$$\rho(\mathbf{x}_i, \mathbf{x}_j) \le M\left(\frac{d\,v_d(R)}{n}\right)^{1/d}.$$

With this condition, we get about $M^d$ neighbors for each node. The following theorem holds.

**Theorem 1.** *Assume that $d = d(n)$ is such that $\log\log n \ll d(n) \ll \log n$ and $R = R(n)$ is such that $e^R \ll n^{\frac{1}{d-1}} \cdot d^{-1/2}$. For the exact NNS, let $M$ be a constant s.t. $M > \sqrt{2}$. Then, with probability $1 - o(1)$, $G(M)$-based NNS finds the exact nearest neighbor. For $c, r$-ANN with some constant $c > 1$, let $M$ be a constant s.t. $M > \sqrt{\frac{4c^2}{3c^2-1}}$. Then, with probability $1 - o(1)$, $G(M)$-based NNS solves $c, r$-ANN for any $r$.*

*In both cases, the time complexity is $O\left(M^d \cdot n^{1/d} \cdot R \cdot (v_d(R))^{-1/d}\right)$;*

*the space complexity is $O\left(M^d \cdot n \cdot \log n\right)$.*

The formal proof of this theorem is given in Appendices A-C, and in Appendix D we prove a similar theorem for $k$-NN graphs with $k = [M^d]$.

Theorem 1 covers several different regimes. First, assume that $R \ll 1/\sqrt{d}$. In this case, the geometry is approximately Euclidean. We have $v_d(R) \sim R^d/d$, so the following corollary holds.

**Corollary 1.** *If $R = R(n) \ll 1/\sqrt{d}$, then in Theorem 1 the time complexity is $O\left(M^d \cdot n^{1/d}\right)$.*

As expected, this result is similar to the corresponding theorem in Prokhorenkova & Shekhovtsov (2020). However, as $R$ becomes larger, time complexity changes. For instance, if $R \gg \log d$, then we have $v_d(R) \sim e^{R(d-1)}\,\Theta(d^{-1}2^{-d})$. Thus, we get the following corollary.

**Corollary 2.** *If $R = R(n) \gg \log d$, then in Theorem 1 we obtain the time complexity $O\left(M^d \cdot n^{1/d} \cdot R \cdot e^{-R(d-1)/d}\right)$.*

From this corollary we see that the time complexity decreases with the radius $R$. Thus, to make the complexity smaller, we should take larger values of $R$. According to Theorem 1, we can take $R$ such that $e^R = n^{\frac{1}{d-1}}/(\sqrt{d}\,\varphi)$, where $\varphi = \varphi(n) \to \infty$. We obtain the following corollary.

**Corollary 3.** *If for an arbitrary $\varphi = \varphi(n) \to \infty$ we have $e^R = n^{\frac{1}{d-1}}/(\sqrt{d}\,\varphi)$ and $R \gg \log d$, then in Theorem 1 we get the time complexity $O\left(M^d \cdot \log n \cdot d^{-1/2} \cdot \varphi\right)$.*

Thus, in the hyperbolic space, the number of steps can be reduced from $n^{1/d}$ to about $\log n$. This shows that compared to the Euclidean case, NNS over nearest neighbor graphs in the hyperbolic space can be more efficient (under the conditions considered in this paper). Intuitively, the difference between Euclidean and hyperbolic regimes is due to the fact that the volume of a hyperbolic ball grows exponentially with its radius. As a consequence, we need fewer steps to reach any element from, e.g., the origin of the ball. Note that in the Euclidean space, we can also reduce the number of steps to $\log n$ via adding long edges (Prokhorenkova & Shekhovtsov, 2020). In contrast, in hyperbolic space, this can be achieved without additional tricks. Our experiments on synthetic datasets confirm that without long edges, NNS is much more efficient in the hyperbolic space when dimension $d$ is small.

Finally, let us discuss the sparse regime with $d \gg \log n$. Here we show that the elements are tightly concentrated near the boundary of the ball and thus the convergence guarantees resemble those on the sphere. In this case, we construct the graph $G(M)$ by connecting such elements $\mathbf{x}_i$ and $\mathbf{x}_j$ that $\cosh \rho(\mathbf{x}_i, \mathbf{x}_j) \leq \cosh^2 R - \sinh^2 R \cdot \sqrt{\frac{2M \log n}{d}}$. To formulate the result, we introduce the following notation: for any constant $c > 1$ let $\alpha_c := \frac{\cosh^2 R - \cosh(\operatorname{arccosh}(\cosh^2 R)/c)}{\sinh^2 R}$.

**Theorem 2.** *Assume that $d \gg \log n$ and $R \gg \log d$. Let $c > 1$ and let $M$ be any constant such that $M < \frac{\alpha_c^2}{\alpha_c^2 + 1}$. Then, with probability $1 - o(1)$, $G(M)$-based NNS solves $c, r$-ANN (for any $r$); the time complexity of this procedure is $\Theta\left(n^{1-M+o(1)}\right)$; the space complexity is $\Theta\left(n^{2-M+o(1)}\right)$.*

The proof is given in Appendix E. This theorem shows that the complexity of hyperbolic NNS for sparse datasets is similar to that on a sphere. However, for large $d$, the calculation error in the hyperbolic space may become significant even for moderate values of $R$. We also remark that the condition $R \gg \log d$ is chosen to simplify the proof using the approximation (3), but this restriction does not seem to be necessary for the theorem to hold.

## 5 EXPERIMENTS

In this section, we first illustrate our theoretical results from Section 4.3 on synthetic datasets uniformly distributed within balls of different radii in the hyperbolic space. Then, we compare graph-based approaches with the algorithm proposed by Wu & Charikar (2020) on Poincaré GloVe word embeddings (Tifrea et al., 2019). Finally, we compare the performance on Euclidean and Poincaré GloVe datasets to show that the efficiency can be even better for hyperbolic representations. Additional experiments can be found in Appendix; the code supplements the submission.

### 5.1 ILLUSTRATING THEORETICAL RESULTS ON SYNTHETIC UNIFORM DATASETS

In this section, we illustrate Theorem 1 and our corollary that for simple nearest-neighbor graphs and sufficiently small dimension $d$, graph-based NNS is more efficient in the hyperbolic space. For this, we follow the setup of Prokhorenkova & Shekhovtsov (2020) and perform a synthetic experiment. We generate datasets uniformly distributed in different spaces: 1) on a sphere (Prokhorenkova & Shekhovtsov, 2020), 2) within a Euclidean ball, 3) within a ball of radius $R$ in the hyperbolic space of curvature $-1$. For all spaces, we fix $n = 10^6$ and vary the dimension $d \in \{2, 4, 6\}$.

For the hyperbolic space, we consider different values of $R$ which is equivalent to varying curvature. Recall that Theorem 1 requires $e^R \ll n^{1/(d-1)}/\sqrt{d}$. Thus, roughly speaking, we obtain the following upper bound on $R$: $R_{max} = \frac{\log n}{d-1} - \log \sqrt{d}$. For $d = 2$, we get $R_{max} \approx 13.5$; for $d = 4$, we get $R_{max} \approx 3.9$; while for $d = 6$ we have $R_{max} \approx 1.9$. Thus, even for $d = 6$, the bound on the radius is sufficiently small and the obtained geometry becomes similar to Euclidean (if we consider the theoretically analyzed parameter range). Thus, we do not consider $d > 6$ in this experiment.

First, we analyze the performance of greedy search over nearest-neighbor graphs; this setup corresponds to our theoretical analysis. To get the complexity-vs-accuracy curves, we vary the graph

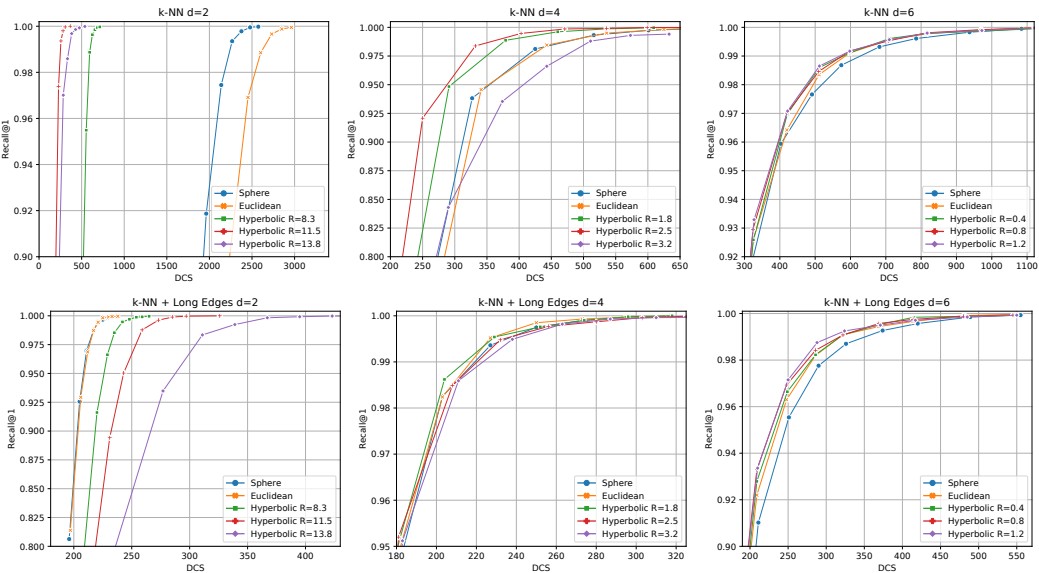

Figure 1: Greedy search over NN graphs without (top row) and with (bottom row) long edges

degree. As a primary measure of the search accuracy, we consider `Recall@1`. The search complexity is the average number of distance computations (DCS) per query. In Figure 1 we see that spherical and Euclidean spaces are similar. Recall that the Euclidean space can be thought of as the hyperbolic one with $R \to 0$. Then, as $R$ increases, the performance of NNS in the hyperbolic space improves. However, after some point, larger values of $R$ reduce the quality of NNS. It is worth noting that this threshold agrees well with the dynamic of $R_{max}$ (while being smaller, as expected).

Then, we analyze what happens if we add long edges known to be essential for NNS in spherical and Euclidean spaces when the dimension $d$ is small. To incorporate long edges into NN graphs, we follow the procedure described in Prokhorenkova & Shekhovtsov (2020). As expected, long edges significantly boost the performance of NNS in spherical and Euclidean spaces. Note that they also improve performance in hyperbolic spaces, but the boost is smaller. As a result, the efficiency of NNS in all spaces becomes similar.

We also conduct similar experiments with greedy search replaced by the best-first search, which is a standard technique for graph-based NNS performance improvement. The conclusions are the same; we include the figures in Appendix F.

## 5.2 COMPARISON WITH BASELINE

Let us analyze the performance of graph-based NNS in a more realistic setup. For this, we collect the largest publicly available Poincaré GloVe vocabulary[1] of $189,533$ unique tokens. Poincaré GloVe was shown to outperform the Euclidean representation for tasks of similarity, analogy, and hypernymy detection (Tifrea et al., 2019). The dimensionality of the embeddings is $100$. The set of tokens is randomly split into the base set of $180K$ elements and the query set of the remaining elements for evaluation. Note that the dataset is not uniform and the dimensionality is sufficiently large, so our theoretical analysis cannot be directly applied.

On the base set, we construct the HNSW graph (Malkov & Yashunin, 2018) that is known to achieve state-of-the-art results on many datasets. As for the search algorithm, we perform the best-first search. The parameter $efSearch$ determines the stopping criteria of the best-first search and allows us to vary the complexity versus accuracy tradeoff. For the baseline, the only practical algorithm we are aware of is proposed by Wu & Charikar (2020). The algorithm is called Spherical Shell: it is suggested to split the Poincaré ball into thin annuli and perform approximate Euclidean NNS (the authors use LSH) for some of them separately. Unfortunately, the implementation is not provided

---

[1] https://github.com/alex-tifrea/poincare_glove

by the authors, so we use our implementation. To make the comparison fair, we count the number of distance computations instead of the actual time, as the latter may depend on the efficiency of our implementation. The algorithm details and parameter tuning are discussed in Appendix F.

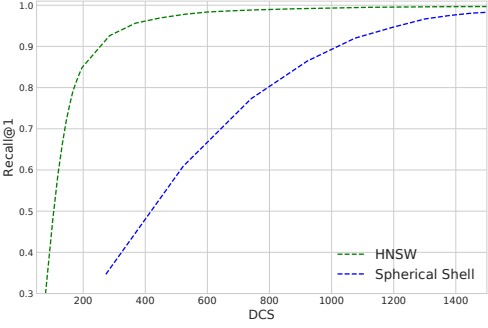
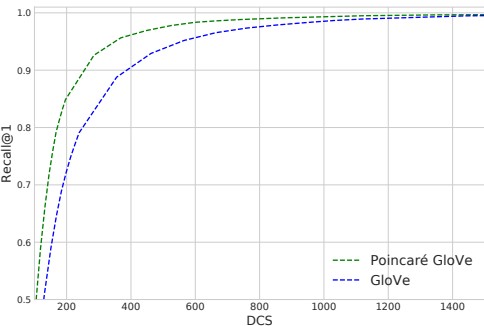

Figure 2: HNSW vs Spherical Shell on Poincaré GloVe embeddings

Figure 3: Efficiency of HNSW in hyperbolic space vs Euclidean space

The results are shown in Figure 2. We see that HNSW significantly outperforms the baseline. This confirms that applying an algorithm directly to the hyperbolic space is more efficient than reducing the problem to Euclidean NNS. Some additional experiments can be found in Appendix: we evaluate another graph-based algorithm and consider more datasets.

## 5.3 EUCLIDEAN VS POINCARÉ GLOVE

We also compare the efficiency of graph-based methods for Euclidean (Pennington et al., 2014) and Poincaré (Tifrea et al., 2019) GloVe embeddings of the same tokens. Note that the results cannot be compared directly: GloVe and Poincaré GloVe have different geometry, so nearest neighbors for the same queries may differ. Thus, we only aim to show that switching from Euclidean to hyperbolic embeddings does not make NNS significantly less efficient. Figure 3 shows that HNSW built on embeddings in the hyperbolic space is even *more efficient* than HNSW over Euclidean embeddings.[2] This further confirms the applicability of graph-based methods to hyperbolic representations.

## 6 CONCLUSION

This paper conducts a theoretical and empirical analysis of graph-based nearest neighbor search in hyperbolic spaces. First, we theoretically analyze the performance of greedy search over nearest-neighbor graphs assuming that the dataset is uniformly distributed over a ball of radius $R$ in the hyperbolic space. Assuming the dense regime ($d \ll \log n$) and that the absolute value of the curvature is not too large, we show that locally graph-based exploration is similar to the Euclidean case. However, globally, hyperbolic geometry allows for reducing the number of steps significantly (for plain NN graphs). For sparse regime ($d \gg \log n$), we show that NNS has similar complexity in Euclidean and hyperbolic spaces.

From a practical perspective, we show that graph-based algorithms are superior to other existing methods. Also, compared to the Euclidean NNS, switching to hyperbolic space does not lead to significantly increased complexity and may even reduce the query time. We hope that our research will attract more attention to theoretical and empirical analysis of NNS in hyperbolic spaces and, in particular, to graph-based methods.

Our work is the first step towards understanding graph-based methods for hyperbolic representations, and there are plenty of directions for future research. In particular, the assumptions of Theorem 1 imply locally Euclidean behavior of the algorithm. It would be useful to extend this theorem beyond locally Euclidean neighborhoods by relaxing the upper bound on $R$. Of course, it is also important to relax the uniformity assumption, but this problem is still to be addressed for a more simple Euclidean case.

---

[2]Note that computing the hyperbolic distance has the same complexity as computing the inner product, as follows from Equation (1).

ACKNOWLEDGMENTS

A part of work of Nikolay Bogachev was done during his postdoc at Skoltech.

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

# A   AUXILIARY GEOMETRICAL RESULTS

In this section, we prove some auxiliary geometrical results that will be useful throughout the proofs. First, we analyze the properties of Euclidean balls and their intersections. Then, we obtain similar results for the hyperbolic space if the ball's radii are sufficiently small.

## A.1   EUCLIDEAN BALLS AND THEIR INTERSECTIONS

For a ball $B = B(1)$ of unit radius in a $d$-dimensional Euclidean space, its volume is

$$\mathrm{Vol}_{\mathbb{E}^d}(B) = \frac{\pi^{\frac{d}{2}}}{\Gamma\left(\frac{d}{2} + 1\right)}.$$

In this section, we estimate volumes of spherical caps and then use the obtained bounds to estimate the volumes of balls' intersections.

Let $B(\mathbf{x}, \alpha)$ be a ball of radius $\alpha$ centered at $\mathbf{x}$, and $C_\alpha(\mathbf{x}, \mathbf{z}, h) \subseteq B(\mathbf{x}, \alpha)$ be a $d$–dimensional spherical cap of height $h$ centered at $\mathbf{z} \in \mathcal{S}_\alpha(\mathbf{x})$, i.e., $C_\alpha(\mathbf{x}, \mathbf{z}, h) = \{\mathbf{y} \in B(\mathbf{x}, \alpha) : \langle \mathbf{y} - \mathbf{x}, \mathbf{z} - \mathbf{x}\rangle \geq \alpha(\alpha - h)\}$. Let $C_\alpha(h) := \mathrm{Vol}_{\mathbb{E}^d}\left(C_\alpha(\mathbf{x}, \mathbf{z}, h)\right)$ be the volume of this spherical cap.

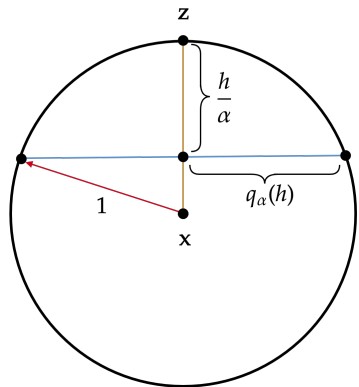

**Definition 1.** *Let* $q_\alpha(h) := \left(1 - \left(1 - \frac{h}{\alpha}\right)^2\right)^{1/2}$ *be the "relative radius" of a spherical cap (see* $q_\alpha(h)$ *in the figure).*

**Lemma 2.** *The volume* $C_\alpha(h)$ *of a spherical cap* $C_\alpha(\mathbf{x}, \mathbf{z}, h)$ *can be estimated as follows:*

$$\Theta(d^{-1}) \leq \frac{C_\alpha(h)}{\mathrm{Vol}_{\mathbb{E}^{d-1}}(B) \cdot \alpha^d \cdot q_\alpha(h)^{d+1}} \leq \Theta(d^{-\frac{1}{2}}).$$

*Proof.* For brevity, let $q = q_\alpha(h)$. Let $\gamma = \arccos(1 - h/\alpha), 0 \leq \gamma \leq \frac{\pi}{2}$. The volume of a spherical cap of height $h$ is known to be:

$$C_\alpha(h) = \mathrm{Vol}_{\mathbb{E}^{d-1}}(B) \cdot \alpha^d \cdot \int_0^\gamma \sin^d(t)\, dt.$$

First, rewrite the integral:

$$\int_0^\gamma \sin^d(t)\, dt = |x = \sin^2 t| = \frac{1}{2} \int_0^{q^2} x^{\frac{d-1}{2}} (1 - x)^{-\frac{1}{2}}\, dx$$

$$= |x = q^2 t| = \frac{1}{2} q^{d+1} \int_0^1 t^{\frac{d-1}{2}} \left(1 - q^2 t\right)^{-\frac{1}{2}}\, dt.$$

To prove the lemma, it remains to show that

$$\Theta(d^{-1}) \leq \int_0^1 t^{\frac{d-1}{2}} \left(1 - q^2 t\right)^{-\frac{1}{2}}\, dt \leq \Theta\left(d^{-\frac{1}{2}}\right).$$

Indeed,

$$\Theta(d^{-1}) = \int_0^1 t^{\frac{d-1}{2}} \, dt \le \int_0^1 t^{\frac{d-1}{2}} \left(1 - q^2 t\right)^{-\frac{1}{2}} \, dt$$

$$\le \int_0^1 t^{\frac{d-1}{2}} \left(1 - t\right)^{-\frac{1}{2}} \, dt = \mathrm{B}\left(\frac{d+1}{2}, \frac{1}{2}\right) = \Theta\left(d^{-\frac{1}{2}}\right),$$

where $\mathrm{B}(\cdot, \cdot)$ denotes the beta function. This concludes the proof. $\qquad \square$

Let us now estimate the volume of the intersection of two Euclidean balls. Let $W_{\mathbf{x}_i, \mathbf{x}_j}(\alpha, \beta)$ denote the intersection of $B(\mathbf{x}_i, \alpha)$ and $B(\mathbf{x}_j, \beta)$, i.e., $W_{\mathbf{x}_i, \mathbf{x}_j}(\alpha, \beta) = \{\mathbf{y} : \rho(\mathbf{x}_i, \mathbf{y}) \le \alpha, \, \rho(\mathbf{x}_j, \mathbf{y}) \le \beta\}$. Let $s := \rho(\mathbf{x}_i, \mathbf{x}_j)$ be the distance between the centers and $W(\alpha, \beta, s) := \mathrm{Vol}_{\mathbb{E}^d}\left(W_{\mathbf{x}_i, \mathbf{x}_j}(\alpha, \beta)\right)$ denote the volume of the intersection. The following lemma estimates the value $W(\alpha, \beta, s)$ under various conditions on $\alpha, \beta, s$.

**Lemma 3.** *W.l.o.g., assume that $\alpha \le \beta$. Define $A := \frac{(\alpha+\beta+s)(\alpha+\beta-s)(s+\alpha-\beta)(s+\beta-\alpha)}{4s^2}$.*

1. *If $\alpha + \beta < s$, then $W(\alpha, \beta, s) = 0$.*

2. *If $s - \alpha \le \beta < \sqrt{\alpha^2 + s^2}$, then*

$$\Theta\left(d^{-1}\right) \le \frac{W(\alpha, \beta, s) \cdot \alpha\beta}{\mathrm{Vol}_{\mathbb{E}^{d-1}}(B) \cdot (\alpha + \beta) \cdot A^{\frac{d+1}{2}}} \le \Theta\left(d^{-\frac{1}{2}}\right). \tag{5}$$

3. *If $\sqrt{\alpha^2 + s^2} \le \beta < s + \alpha$, then*

$$\frac{1}{2} \le \frac{W(\alpha, \beta, s)}{\mathrm{Vol}_{\mathbb{E}^d}(B) \cdot \alpha^d} \le 1, \tag{6}$$

$$\frac{\mathrm{Vol}_{\mathbb{E}^d}(B(\alpha)) - W(\alpha, \beta, s)}{\mathrm{Vol}_{\mathbb{E}^{d-1}}(B) \cdot A^{\frac{d+1}{2}} \cdot \alpha^{-1}} \le \Theta\left(d^{-\frac{1}{2}}\right). \tag{7}$$

4. *If $s + \alpha \le \beta$, then*

$$W(\alpha, \beta, s) = \mathrm{Vol}_{\mathbb{E}^d}(B) \cdot \alpha^d. \tag{8}$$

*Proof.* Let $h_\alpha := \frac{\beta^2 - (s-\alpha)^2}{2s}$, $h_\beta := \frac{\alpha^2 - (s-\beta)^2}{2s}$, $\widetilde{h}_\alpha := \frac{(\alpha+s)^2 - \beta^2}{2s}$.

1. If $\alpha + \beta < s$, then the balls do not intersect and thus $W(\alpha, \beta, s) = 0$, see Figure 4a.

2. If $s - \alpha \le \beta < \sqrt{\alpha^2 + s^2}$, then, according to Figure 4b,

$$W(\alpha, \beta, s) = C_\alpha(h_\alpha) + C_\beta(h_\beta).$$

By Lemma 2, we have

$$\Theta\left(d^{-1}\right) \le \frac{W(\alpha, \beta, s)}{\mathrm{Vol}_{\mathbb{E}^{d-1}}(B) \cdot \left(\alpha^d q_\alpha^{d+1}(h_\alpha) + \beta^d q_\beta^{d+1}(h_\beta)\right)} \le \Theta\left(d^{-\frac{1}{2}}\right).$$

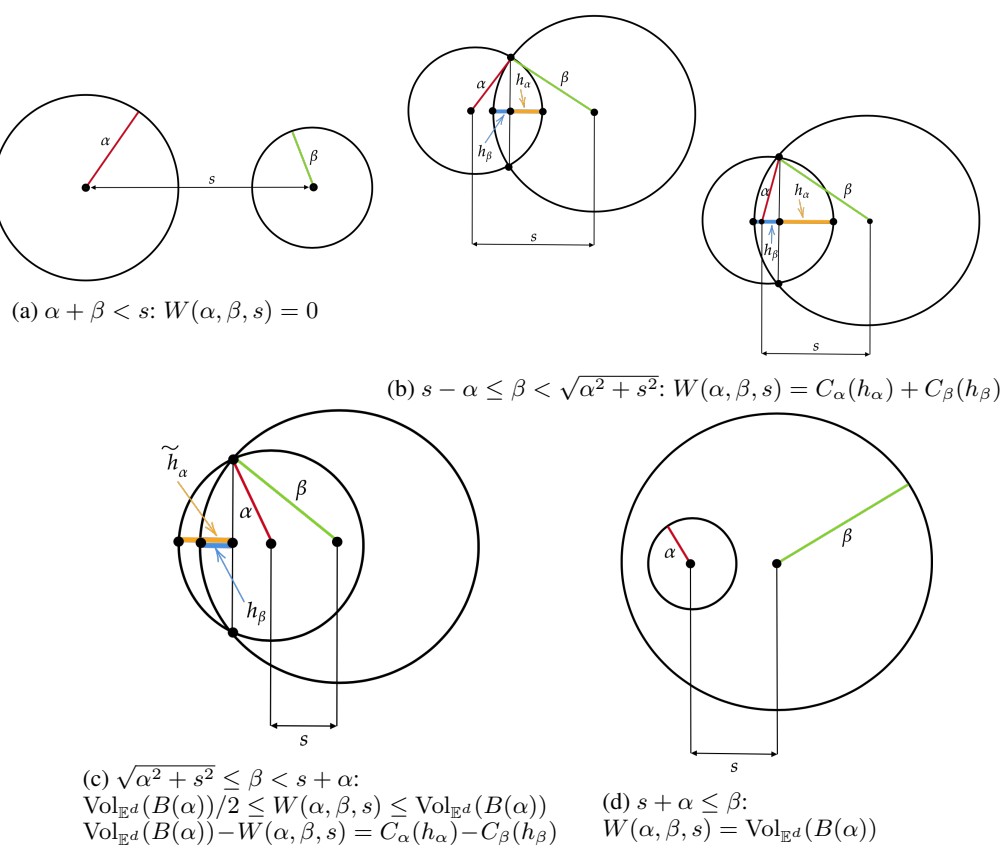

(a) $\alpha + \beta < s$: $W(\alpha, \beta, s) = 0$

(b) $s - \alpha \leq \beta < \sqrt{\alpha^2 + s^2}$: $W(\alpha, \beta, s) = C_\alpha(h_\alpha) + C_\beta(h_\beta)$

(c) $\sqrt{\alpha^2 + s^2} \leq \beta < s + \alpha$:
$\mathrm{Vol}_{\mathbb{E}^d}(B(\alpha))/2 \leq W(\alpha, \beta, s) \leq \mathrm{Vol}_{\mathbb{E}^d}(B(\alpha))$
$\mathrm{Vol}_{\mathbb{E}^d}(B(\alpha)) - W(\alpha, \beta, s) = C_\alpha(h_\alpha) - C_\beta(h_\beta)$

(d) $s + \alpha \leq \beta$:
$W(\alpha, \beta, s) = \mathrm{Vol}_{\mathbb{E}^d}(B(\alpha))$

Figure 4: Different cases in the proof of Lemma 3

Substituting $q_\alpha$, $q_\beta$, $h_\alpha$, $h_\beta$, we obtain that

$$\alpha^d q_\alpha^{d+1}(h_\alpha) + \beta^d q_\beta^{d+1}(h_\beta)$$

$$= \alpha^d \left( \frac{\alpha^2 - (\alpha - h_\alpha)^2}{\alpha^2} \right)^{\frac{d+1}{2}} + \beta^d \left( \frac{\beta^2 - (\beta - h_\beta)^2}{\beta^2} \right)^{\frac{d+1}{2}}$$

$$= \frac{1}{\alpha} \left( h_\alpha(2\alpha - h_\alpha) \right)^{\frac{d+1}{2}} + \frac{1}{\beta} \left( h_\beta(2\beta - h_\beta) \right)^{\frac{d+1}{2}}$$

$$= \frac{1}{\alpha} \left( \frac{(\beta^2 - (s-\alpha)^2)((s+\alpha)^2 - \beta^2)}{4s^2} \right)^{\frac{d+1}{2}} + \frac{1}{\beta} \left( \frac{(\alpha^2 - (s-\beta)^2)((s+\beta)^2 - \alpha^2)}{4s^2} \right)^{\frac{d+1}{2}}$$

$$= \left( \frac{1}{\alpha} + \frac{1}{\beta} \right) A^{\frac{d+1}{2}}.$$

3. If $\sqrt{\alpha^2 + s^2} \le \beta < s + \alpha$, then (see Figure 4c)

$$\frac{\mathrm{Vol}_{\mathbb{E}^d}(B(\alpha))}{2} \le W(\alpha, \beta, s) \le \mathrm{Vol}_{\mathbb{E}^d}(B(\alpha)),$$

Also, according to Figure 4c,

$$\mathrm{Vol}_{\mathbb{E}^d}(B(\alpha)) - W(\alpha, \beta, s) = C_\alpha(\widetilde{h}_\alpha) - C_\beta(h_\beta) \le C_\alpha\left(\widetilde{h}_\alpha\right).$$

By Lemma 2, we have

$$\frac{\mathrm{Vol}_{\mathbb{E}^d}(B) \cdot \alpha^d - W(\alpha, \beta, s)}{\mathrm{Vol}_{\mathbb{E}^{d-1}}(B) \cdot \alpha^d \cdot q_\alpha^{d+1}(\widetilde{h}_\alpha)} \le \Theta\left(d^{-\frac{1}{2}}\right).$$

Substituting $q_\alpha$ and $\widetilde{h}_\alpha$, we get

$$\alpha^d q_\alpha^{d+1}(\widetilde{h}_\alpha) = \alpha^d \left( \frac{\alpha^2 - (\alpha - \widetilde{h}_\alpha)^2}{\alpha^2} \right)^{\frac{d+1}{2}} = \frac{1}{\alpha} \left( \widetilde{h}_\alpha(2\alpha - \widetilde{h}_\alpha) \right)^{\frac{d+1}{2}}$$

$$= \frac{1}{\alpha} \left( \frac{((s+\alpha)^2 - \beta^2)(\beta^2 - (s-\alpha)^2)}{4s^2} \right)^{\frac{d+1}{2}} = \frac{1}{\alpha} A^{\frac{d+1}{2}}.$$

4. If $s + \alpha \le \beta$, then $W(\alpha, \beta, s) = \mathrm{Vol}_{\mathbb{E}^d}(B(\alpha)) = \mathrm{Vol}_{\mathbb{E}^d}(B) \cdot \alpha^d$, see Figure 4d.

□

The following lemma will also be helpful in our proofs.

**Lemma 4.** $W(\alpha, \beta, \alpha)$ and $W(\beta, \alpha, \alpha)$ are non-decreasing functions of $\alpha$.

*Proof.* Let us consider the configuration of balls shown in Figure 5. The $d$-dimensional ball centered at $\mathbf{q}'$ is contained within the ball centered at $\mathbf{q}$. Thus, the intersection $\mathbf{A}'\mathbf{B}'\mathbf{x}$ of the balls centered at $\mathbf{x}$ and $\mathbf{q}'$ is contained in the intersection $\mathbf{ABx}$ of the balls centered at $\mathbf{x}$ and $\mathbf{q}$. Clearly, the center $\mathbf{q}'$ of the ball which is closer to $\mathbf{x}$ can be taken on the geodesic between $\mathbf{q}$ and $\mathbf{x}$ without loss of generality. The same argument works when the ball centered at $\mathbf{x}$ contains the centers $\mathbf{q}$, $\mathbf{q}'$ of the other balls. □

## A.2 HYPERBOLIC VOLUMES

First, we briefly recall the basic properties of hyperbolic space and then analyze hyperbolic volumes and their Euclidean counterparts. On a small scale, hyperbolic volumes behave in an almost Euclidean way, while their behavior is fundamentally different on a larger scale.

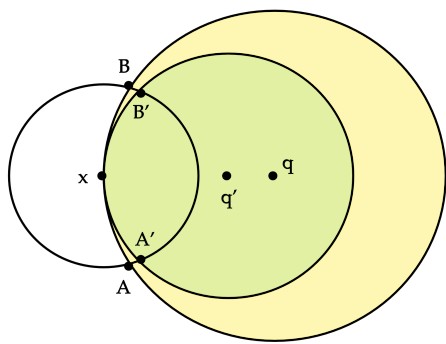

Figure 5: Monotonicity of $W(\alpha, \beta, \alpha)$ and $W(\beta, \alpha, \alpha)$

### A.2.1  BACKGROUND ON HYPERBOLIC SPACE

Recall that by $\mathbb{R}^{d,1}$ we denote the $(d+1)$-dimensional *Minkowski space*, i.e., the real vector space $\mathbb{R}^{d+1}$ equipped with the inner product of signature $(d, 1)$:

$$\langle \mathbf{x}, \mathbf{y} \rangle_h = -x_0 y_0 + x_1 y_1 + \ldots + x_d y_d.$$

The *vector model* of the *d-dimensional hyperbolic Lobachevsky space* $\mathbb{H}^d$ is the connected component of the standard two-sheeted hyperboloid contained in the *future light cone*:

$$\mathbb{H}^d = \{\mathbf{x} \in \mathbb{R}^{d,1} \mid \langle \mathbf{x}, \mathbf{x} \rangle_h = -1, x_0 > 0\}. \tag{9}$$

The hyperbolic metric is given by

$$\cosh \rho(\mathbf{x}, \mathbf{y}) = -\langle \mathbf{x}, \mathbf{y} \rangle_h.$$

The coordinates $(x_0, x_1, \ldots, x_d)$ satisfy the following system of equations:

$$\begin{aligned}
x_0 &= \cosh(t_1), \\
x_1 &= \sinh(t_1) \cos(t_2), \\
&\quad \ldots \\
x_{d-1} &= \sinh(t_1) \sin(t_2) \ldots \sin(t_{d-1}) \cos(t_d), \\
x_d &= \sinh(t_1) \sin(t_2) \ldots \sin(t_{d-1}) \sin(t_d).
\end{aligned}$$

This hyperbolic coordinate parameterization of $\mathbb{H}^d$ is the map

$$\chi \colon [0, +\infty) \times [0, \pi]^{d-2} \times [0, 2\pi] \to \mathbb{H}^d$$

defined by

$$\chi(t_1, \ldots, t_d) = (x_0, x_1, \ldots, x_d).$$

A subset $U \subset \mathbb{H}^d$ is called measurable if $\chi^{-1}(U)$ is measurable with respect to the standard Lebesgue measure on $\mathbb{R}^d$. Let $U \subset \mathbb{H}^d$ be a measurable set. Then, the hyperbolic volume of $U$ is defined by

$$\mathrm{Vol}_{\mathbb{H}^d}(U) = \int_{\chi^{-1}(U)} \sinh^{d-1}(t_1) \sin^{d-2}(t_2) \ldots \sin(t_{d-1}) \, dt_1 dt_2 \ldots dt_{d-1} dt_d.$$

### A.2.2  APPROXIMATING SMALL HYPERBOLIC VOLUMES BY THEIR EUCLIDEAN COUNTERPARTS

In this section, we show that small hyperbolic volumes can be approximated by Euclidean ones. This is very helpful for our analysis and will allow us to obtain the analogs of Lemma 2 and 3 for small hyperbolic balls.

From the definition of the hyperbolic space (9), we see that $x_0 = \sqrt{1 + x_1^2 + \ldots + x_d^2}$. Let $U \subset \mathbb{H}^d$ be a measurable set containing the point $\mathbf{z} = (1, 0, \ldots, 0)$, and let $U'$ be the projection of $U$ along the $x_0$–coordinate from the hyperboloid onto the tangent space $T_{\mathbf{z}}\mathbb{H}^d = \{\mathbf{x} \in \mathbb{R}^{d,1} \mid x_0 = 1\} \simeq \mathbb{R}^d$ realised by the map $p : \mathbb{H}^d \to \mathbb{R}^d$. Let $\hat{c}$ be the central projection $\hat{c} : \mathbb{H}^d \to T_{\mathbf{z}}\mathbb{H}^d \simeq \mathbb{R}^d$ from the origin $\mathbf{o} = (0, \ldots, 0)$. Let $F : U' \to \hat{c}U)$ be the map that satisfies $\hat{c} = F \circ p$.

Let us note that the central projection $\hat{c}$ maps balls centered at $\mathbf{z}$ in $\mathbb{H}^d$ to balls centered at $p(\mathbf{z}) = \mathbf{z}$ in $\mathbb{R}^d$, and also spherical caps of such balls in $\mathbb{H}^d$ have spherical caps in $\mathbb{R}^d$ as their respective images, since every spherical cap is an intersection of some ball with a half-space, which is bounded by some hyperplane in $\mathbb{R}^{d,1}$ (the central projection of this hyperplane is also a hyperplane in $\mathbb{R}^d$).

According to Ratcliffe (2019),

$$\mathrm{Vol}_{\mathbb{H}^d}(U) = \int_{U'} \frac{dx_1 \ldots dx_d}{\sqrt{1 + x_1^2 + \ldots + x_d^2}}.$$

We need to compare $\mathrm{Vol}_{\mathbb{H}^d}(U)$ to $\mathrm{Vol}_{\mathbb{E}^d}(\hat{c}U))$ in order to conclude that these quantities approximate each other in some appropriate regime.

**Lemma 5.** *Suppose that $U \subset \mathbb{H}^d$ is a measurable set such that $U \subset B(\mathbf{z}, r)$. If $r \ll \frac{1}{\sqrt{d}}$ and $d \to \infty$, then we have*

$$\mathrm{Vol}_{\mathbb{H}^d}(U) = \mathrm{Vol}_{\mathbb{E}^d}(\hat{c}U)) \left(1 - O\left(d\, r^2\right)\right).$$

*Proof.* We have that $U' = p(U) \subset B_{\mathbb{R}^d}(0, \sinh(r))$, hence $x_1^2 + \ldots + x_d^2 \leq \sinh^2(r)$. Then,

$$\int_{U'} \frac{dx_1 \ldots dx_d}{\cosh(r)} < \mathrm{Vol}_{\mathbb{H}^d}(U) = \int_{U'} \frac{dx_1 \ldots dx_d}{\sqrt{1 + x_1^2 + \ldots + x_d^2}} < \int_{U'} dx_1 \ldots dx_d = \mathrm{Vol}_{\mathbb{E}^d}(U').$$

By using Taylor series expansion, we obtain

$$\int_{U'} \frac{dx_1 \ldots dx_d}{\cosh(r)} = \frac{1}{\cosh(r)} \mathrm{Vol}_{\mathbb{E}^d}(U') = \left(1 - \frac{r^2}{2} + \Theta\left(r^4\right)\right) \mathrm{Vol}_{\mathbb{E}^d}(U'),$$

and thus

$$\left(1 - \frac{r^2}{2} + \Theta\left(r^4\right)\right) \mathrm{Vol}_{\mathbb{E}^d}(U') < \mathrm{Vol}_{\mathbb{H}^d}(U) < \mathrm{Vol}_{\mathbb{E}^d}(U').$$

Now let us show that

$$\mathrm{Vol}_{\mathbb{E}^d}(U') = \mathrm{Vol}_{\mathbb{E}^d}(\hat{c}U)) \left(1 - O\left(dr^2\right)\right).$$

Indeed, the map $F : U' \to \hat{c}U$ satisfies $\hat{c} = F \circ p$ and is given by $F = (F_1, \ldots, F_d)$ with

$$F_i : x_i \mapsto \frac{x_i}{\sqrt{1 + x_1^2 + \ldots + x_d^2}}, \quad \text{for } i = 1, \ldots, d,$$

and thus the Jacobian of $F$ satisfies

$$\det \left(\frac{\partial F_i}{\partial x_j}\right)_{i,j=1}^d = \frac{1}{(1 + x_1^2 + \ldots + x_d^2)^{\frac{d+2}{2}}} = 1 - O\left(dr^2\right),$$

whenever $x_1^2 + \ldots + x_d^2 \leq \sinh^2(r)$ and $dr^2 \to 0$. One can conveniently use spherical coordinates in order to perform the respective computation. The lemma now follows from standard Taylor series techniques. □

Let $B_d(\mathbf{x}, r)$ be a closed hyperbolic ball centered at a point $\mathbf{x} \in \mathbb{H}^d$ of radius $r > 0$. Then we can compute its volume as (Ratcliffe, 2019):

$$\mathrm{Vol}_{\mathbb{H}^d}(B_d(\mathbf{x}, r)) = \mathrm{Vol}\left(\mathbb{S}^{d-1}\right) \int_0^r \sinh^{d-1}(t)\, dt.$$

For $r \gg \log d$, we have

$$\mathrm{Vol}_{\mathbb{H}^d}(B_d(\mathbf{x}, r)) \sim \mathrm{Vol}(\mathbb{S}^{d-1}) \exp(r(d-1))\, \Theta(d^{-1}2^{-d}).$$

For $r \ll 1/\sqrt{d}$, we have from Lemma 5 that

$$\mathrm{Vol}_{\mathbb{H}^d}(B_d(\mathbf{x}, r)) \sim \mathrm{Vol}_{\mathbb{E}^d}(B_d(\mathbf{x}, r)) \sim \mathrm{Vol}(\mathbb{S}^{d-1})\, d^{-1}\, r^d\, \Theta(1).$$

### A.2.3 SMALL HYPERBOLIC BALLS AND THEIR INTERSECTIONS

Lemma 5 allows us to transfer our results for Euclidean balls and their intersections to the hyperbolic space when the radii of the balls are sufficiently small. Let $C_\alpha(h)$ be the volume of a spherical cap of height $h$ in the hyperbolic space. The following lemma follows from Lemma 2 and Lemma 5.

**Lemma 6.** *For $\alpha \ll 1/\sqrt{d}$, the volume of a hyperbolic spherical cap $C_\alpha(h)$ can be estimated as follows:*

$$\Theta(d^{-1}) \le \frac{C_\alpha(h)}{\mathrm{Vol}_{\mathbb{E}^{d-1}}(B) \cdot \alpha^d \cdot q_\alpha(h)^{d+1}} \le \Theta(d^{-\frac{1}{2}}).$$

Let us now estimate the volume of the intersection of two hyperbolic balls. As for the Euclidean case, denote by $W_{\mathbf{x}_i,\mathbf{x}_j}(\alpha,\beta)$ the intersection of $B(\mathbf{x}_i,\alpha)$ and $B(\mathbf{x}_j,\beta)$, i.e., $W_{\mathbf{x}_i,\mathbf{x}_j}(\alpha,\beta) = \{\mathbf{y} : \rho(\mathbf{x}_i,\mathbf{y}) \le \alpha, \rho(\mathbf{x}_j,\mathbf{y}) \le \beta\}$. Let $s := \rho(\mathbf{x}_i,\mathbf{x}_j)$ be the distance between the centers and $W(\alpha,\beta,s) := \mathrm{Vol}_{\mathbb{H}^d}\left(W_{\mathbf{x}_i,\mathbf{x}_j}(\alpha,\beta)\right)$ denote the volume of the intersection. The following lemma estimates the value $W(\alpha,\beta,s)$ for various relationships between $\alpha,\beta,s$.

**Lemma 7.** *Assume that $\alpha \le \beta \ll 1/\sqrt{d}$. Define $A := \frac{(\alpha+\beta+s)(\alpha+\beta-s)(s+\alpha-\beta)(s+\beta-\alpha)}{4s^2}$.*

1. *If $\alpha + \beta < s$, then $W(\alpha,\beta,s) = 0$.*

2. *If $s - \alpha \le \beta < \sqrt{\alpha^2 + s^2}$, then*

$$\Theta\left(d^{-1}\right) \le \frac{W(\alpha,\beta,s) \cdot \alpha\beta}{\mathrm{Vol}_{\mathbb{E}^{d-1}}(B) \cdot (\alpha+\beta) \cdot A^{\frac{d+1}{2}}} \le \Theta\left(d^{-\frac{1}{2}}\right). \tag{10}$$

3. *If $\sqrt{\alpha^2 + s^2} \le \beta < s + \alpha$, then*

$$\frac{1}{2} \le \frac{W(\alpha,\beta,s)}{\mathrm{Vol}_{\mathbb{E}^d}(B) \cdot \alpha^d \cdot (1+o(1))} \le 1, \tag{11}$$

$$\frac{\mathrm{Vol}_{\mathbb{H}^d}(B(\alpha)) - W(\alpha,\beta,s)}{\mathrm{Vol}_{\mathbb{E}^{d-1}}(B) \cdot A^{\frac{d+1}{2}} \cdot \alpha^{-1}} \le \Theta\left(d^{-\frac{1}{2}}\right). \tag{12}$$

4. *If $s + \alpha \le \beta$, then*

$$W(\alpha,\beta,s) = \mathrm{Vol}_{\mathbb{H}^d}(B(\alpha)) \sim \mathrm{Vol}_{\mathbb{E}^d}(B) \cdot \alpha^d. \tag{13}$$

*Proof.* To prove this lemma, we follow the proof of Lemma 3, and use Lemma 6 instead of Lemma 2. $\square$

The following lemma is an analog of Lemma 4. Indeed, the argument uses only monotonicity of volumes under set-theoretic inclusions, which holds for both Euclidean and hyperbolic spaces.

**Lemma 8.** *In the hyperbolic space, $W(\alpha,\beta,\alpha)$ and $W(\beta,\alpha,\alpha)$ are non-decreasing functions of $\alpha$.*

## B PROOF OVERVIEW FOR THEOREM 1

Recall that we assume a uniform distribution of the dataset over a $d$-dimensional ball $\mathcal{B}$ of radius $R = R(n)$ such that

$$e^R \ll \frac{n^{1/(d-1)}}{d^{1/2}}. \tag{14}$$

Then, the graph $G(M)$ is constructed using a threshold $\alpha_M = M\delta$ with

$$\delta = \left(\frac{dv_d(R)}{n}\right)^{1/d}. \tag{15}$$

Let us show that under the conditions of the theorem, we have $\delta \ll 1/\sqrt{d}$. First, note that

$$v_d(R) = \int\limits_0^R \sinh^{d-1}(t)\, dt \le \int\limits_0^R \frac{e^{t(d-1)}}{2^{d-1}}\, dt \le \frac{e^{R(d-1)}}{2^{d-1}(d-1)}.$$

Thus, we have

$$\delta \le \left( \frac{d e^{R(d-1)}}{2^{d-1}(d-1)n} \right)^{1/d} = \Theta\left( e^{R(d-1)/d} n^{-1/d} \right) \ll 1/\sqrt{d}.$$

The last inequality is due to (14). Thus, we can apply Lemma 7 throughout the proof.

To prove that the algorithm succeeds w.h.p., we need to guarantee that it does not stop in a local optimum until we find the exact (or approximate) nearest neighbor of the query $\mathbf{q}$. Given a point $\mathbf{x}$ such that $\rho(\mathbf{q}, \mathbf{x}) = \alpha_s$, the probability of making a step towards $\mathbf{q}$ is determined by $W(\alpha_s, \alpha_M, \alpha_s)$: the distance between the points is $\alpha_s$, the neighbors of $\mathbf{x}$ are within the ball of radius $\alpha_M$ centered at $\mathbf{x}$, and we accept those neighbors that are closer to $\mathbf{q}$ than $\mathbf{x}$, i.e., within a ball of radius $\alpha_s$ centered at $\mathbf{q}$. Lemma 9 analyzes the asymptotic behavior of $W(\alpha_s, \alpha_M, \alpha_s)$.

Lemma 4 states that $W(\alpha, \beta, \alpha)$ is a non-increasing function of $\alpha$. Informally, it means that it is easier to make steps towards the query when $\rho(\mathbf{x}, \mathbf{q})$ is larger. In contrast, when $\mathbf{x}$ becomes closer to $\mathbf{q}$, the probability of getting stuck in a local optimum increases. Therefore, to obtain lower bounds on $W(\alpha_s, \alpha_M, \alpha_s)$, we can consider only the case when $\alpha_s$ is small. Below we assume that $\alpha_s = s\delta$, where $s$ is some constant.

Lemma 11 gives the following lower bounds on $W(\alpha_s, \alpha_M, \alpha_s)$ :

1. if $M < s\sqrt{2}$, then $\frac{W(\alpha_s, \alpha_M, \alpha_s)}{V_d(R)} \ge \frac{1}{n} \cdot \left( M^2 - \frac{M^4}{4s^2} \right)^{\frac{d}{2}+o(d)}$;

2. if $M \ge s\sqrt{2}$, then $\frac{W(\alpha_s, \alpha_M, \alpha_s)}{V_d(R)} \ge \frac{1}{n} \cdot s^{d+o(d)}$.

Let us consider one step of the algorithm. We can move forward w.h.p. if $W(\alpha_s, \alpha_M, \alpha_s) \gg 1/n$. Formally speaking, it follows from Corollary 4 of Lemma 11 that if $M$ and $s$ satisfy

$$s > 1 \ \text{ and either } M \ge s\sqrt{2} \ \text{ or } \ M^2 - M^4/4s^2 > 1, \tag{16}$$

then there exists a constant $S > 1$ such that we can make a step towards $\mathbf{q}$ with probability

$$1 - \left( 1 - \frac{S^{d+o(d)}}{n} \right)^{n-1} = 1 - O\left( e^{-S^{d+o(d)}} \right).$$

We make steps of the greedy algorithm towards the query $\mathbf{q}$ as long as the probability of making the next step is large. Below we also prove that the overall probability of failure (taking into account all steps of the algorithm) is small. Now, let us show that under the conditions of the theorem, we reach some neighborhood of $\mathbf{q}$ and then either find the exact nearest neighbor or the approximate near neighbor by making one more step.

First, we consider the exact NNS. In this case, we assume that the nearest neighbor is located within a ball of radius $r\delta$, $0 < r \le 1$, centered at $\mathbf{q}$. Our goal is to find the exact nearest neighbor. According to the statement of the theorem, we have $M > \sqrt{2}$, which implies

$$\frac{M}{\sqrt{2}} > \frac{M^2}{2\sqrt{M^2 - 1}}. \tag{17}$$

While we move towards the query, $s$ becomes smaller. Let us figure out up to which $s$ we can make steps. According to (16), we need either $1 < s \le M/\sqrt{2}$ or $s > M^2/2\sqrt{M^2 - 1}$. According to (17), it reduces to $s > 1$. In other words, we can move forward to any $s > 1$. Then, according to Lemma 14, it is sufficient to have $M^2 > s^2 + r^2$ to reach the exact nearest neighbor of $\mathbf{q}$ in one step. Thus, we can take any $s$ s.t. $1 < s < \sqrt{M^2 - r^2}$: we can reach this distance with greedy search and then jump directly to the nearest neighbor with high probability.

Now, consider the approximate near neighbor search $c, r$-ANN. In this case, we need to find an element that is within a distance $rc$ from $\mathbf{q}$. According to the statement of the theorem, we have $M > \sqrt{4c^2/(3c^2-1)}$.

Note that $F(r) := 2r^2c^2\left(1 - \sqrt{1 - 1/r^2c^2}\right)$ decreases with $r$ for $r > 1/c$ and equals 2 at $r = 1/c$, while $G(r) := \frac{2}{3}\left(r^2 + 1 + \sqrt{r^4 - r^2 + 1}\right)$ increases with $r$ and equals 2 at $r = 1$. Both $F(r)$ and $G(r)$ attain $4c^2/(3c^2 - 1)$ at $r_0 = \sqrt{4c^2/(c^2+1)(3c^2-1)}$. Therefore, we either have

$$1 \geq r > r_0 \quad \text{and} \quad M^2 > F(r) \tag{18}$$

or

$$r_0 \geq r > \frac{1}{c} \quad \text{and} \quad M^2 > G(r). \tag{19}$$

To solve $c, r$-ANN, either we can reach $s$ such that $rc > s$ or we can reach $s$ such that at the next step we can find the exact nearest neighbor. Let us consider the first possibility. Note that

$$s > 1 \text{ and } M^2 > 2s^2\left(1 - \sqrt{1 - \frac{1}{s^2}}\right) \tag{20}$$

is equivalent to (16). In turn, since $2s^2(1 - \sqrt{1 - 1/s^2})$ is a decreasing function of $s$ on $s > 1$, we get that (18) implies the existence of $s$ such that $rc > s > 1$ and

$$M^2 > 2s^2\left(1 - \sqrt{1 - \frac{1}{s^2}}\right) > F(r).$$

Hence, since (16) holds, we can move forward to such $s$. Let us consider the second possibility. Observe that (19) is equivalent to

$$M^2 - r^2 > \frac{M^4}{4(M^2 - 1)}.$$

Therefore, there exists $s$ such that the condition $s^2 < M^2 - r^2$ of Lemma 14 and (16) hold. Since (16) holds, we can move forward to such $s$ and reach the exact nearest neighbor in one step as the condition of Lemma 14 is satisfied.

Above, we show that under the conditions of the theorem, each step of the greedy search is successful with probability $1 - O\left(e^{-S^{d+o(d)}}\right)$. To formally prove that the algorithm succeeds with high probability, we need to limit the number of steps and estimate the overall probability of failure. For this, we show that at each step of the algorithm, we become closer to the query by some fixed positive value. In Lemma 12, we prove that each step reduces the distance from $\mathbf{q}$ by at least $\varepsilon\delta$, where $\varepsilon$ is some positive constant. Clearly, we start at a distance $O(R)$ from $q$. Hence, we are able to bound the number of steps by $O(R/\varepsilon\delta)$ which is $O\left(Rn^{1/d}(v_d(R))^{-1/d}\right)$.

To estimate the overall success probability, we have to consider the dependence of consecutive steps of the algorithm. Lemma 13 implies that consecutive steps are "almost independent" and the overall success probability is $1 - O\left(Rn^{1/d}(v_d(R))^{-1/d}e^{-S^{d+o(d)}}\right)$. The assumption $d \gg \log\log n$ and the fact that $v_d(R) = \Omega(R^d)$ yield $O\left(Rn^{1/d}(v_d(R))^{-1/d}e^{-S^{d+o(d)}}\right) = O\left(e^{\frac{\log n}{d} - S^{d+o(d)}}\right) = o(1)$.

Also, while most of the analysis does not consider the cases when $\alpha_M$-neighborhoods of the dataset elements intersect the boundary $\partial\mathcal{B}$, we need to show that possible intersections do not affect the result. This is addressed in Lemma 15.

Finally, let us discuss the time and space complexity of the algorithm. Let $v$ be a fixed arbitrary node of $G(M)$ and let $N(v)$ denote the (random) number of its neighbors in $G(M)$. Let $f = f(n) = (n-1)\mathrm{Vol}_{\mathbb{H}^d}\left(B(\alpha_M)\right)/\mathrm{Vol}_{\mathbb{H}^d}\left(\mathcal{B}\right)$ denote the upper bound for the expected number of neighbors of $v$ (this is an upper bound due to possible boundary effects). In Lemma 16, we show that $N(v) \leq 3f/2$ with probability at least $1 - O(1/f)$.

To obtain the time complexity of graph-based NNS, we sum up the complexity of all algorithm's steps. In Lemma 17, we show that if $l$ steps of the NNS are made, then the time complexity is $O\left(lfd\right)$ with probability $1 - O(1/lf)$.

In Lemma 18, we show that the number of edges in the graph is $E(G) \leq 3fn/4$ with probability $1 - O(1/fn)$. If we store the graph as the adjacency lists, then the space complexity is $O(E(G) \cdot \log n) = O(fn \log n)$.

Due to the choice of $\delta$, we have $f = O(M^d)$. Hence, w.h.p., the space complexity is $O(n \cdot \log n \cdot M^d)$ and the time complexity is $O\left(R \cdot n^{1/d} \cdot (v_d(R))^{-1/d} \cdot M^d\right)$.

## C  RESULTS USED THROUGHOUT THE PROOF

In this section, we formulate supplemental statements used in the proof. We need these results for the hyperbolic space, but they are also true for the Euclidean one. Thus, we do not specify a particular space in $W(\cdot, \cdot, \cdot)$.

### C.1  ANALYSIS OF CONVERGENCE

Recall that $\alpha_M = M\delta$ and $\alpha_s = s\delta$, where $M$ and $s$ are some constants and $\delta$ is defined in (15). Recall that $\delta \ll 1/\sqrt{d}$.

**Lemma 9.** *Define* $A := \frac{\alpha_M^2 (4\alpha_s^2 - \alpha_M^2)}{4\alpha_s^2}$.

*If* $\alpha_M \leq \alpha_s \sqrt{2}$, *then*

$$\Theta\left(d^{-1}\right) \leq \frac{W(\alpha_s, \alpha_M, \alpha_s) \cdot \alpha_M \alpha_s}{\mathrm{Vol}_{\mathbb{E}^{d-1}}(B) \cdot (\alpha_M + \alpha_s) \cdot A^{\frac{d+1}{2}}} \leq \Theta\left(d^{-\frac{1}{2}}\right). \tag{21}$$

*If* $\alpha_M > \alpha_s \sqrt{2}$, *then*

$$\frac{W(\alpha_s, \alpha_M, \alpha_s)}{\mathrm{Vol}_{\mathbb{E}^d}(B) \alpha_s^d} = \Theta(1). \tag{22}$$

*Proof.* We apply Lemma 7. First, consider the case $\alpha_M \leq \alpha_s$. Then, the case 2 applies to $W(\alpha_M, \alpha_s, \alpha_s)$. From (10) we obtain

$$\Theta\left(d^{-1}\right) \leq \frac{W(\alpha_s, \alpha_M, \alpha_s) \cdot \alpha_M \alpha_s}{\mathrm{Vol}_{\mathbb{E}^{d-1}}(B) \cdot (\alpha_M + \alpha_s) \cdot A^{\frac{d+1}{2}}} \leq \Theta\left(d^{-\frac{1}{2}}\right). \tag{23}$$

Now, consider the case $\alpha_s < \alpha_M$.

1. If $\alpha_s < \alpha_M \leq \alpha_s \sqrt{2}$, then from (10) we obtain

$$\Theta\left(d^{-1}\right) \leq \frac{W(\alpha_s, \alpha_M, \alpha_s) \cdot \alpha_M \alpha_s}{\mathrm{Vol}_{\mathbb{E}^{d-1}}(B) \cdot (\alpha_M + \alpha_s) \cdot A^{\frac{d+1}{2}}} \leq \Theta\left(d^{-\frac{1}{2}}\right).$$

2. If $\alpha_s \sqrt{2} < \alpha_M < 2\alpha_s$, then (11) yields

$$W(\alpha_s, \alpha_M, \alpha_s) = \Theta\left(\mathrm{Vol}_{\mathbb{E}^d}(B) \cdot \alpha_s^d\right).$$

3. If $2\alpha_s \leq \alpha_M$, then (13) gives

$$W(\alpha_s, \alpha_M, \alpha_s) = \mathrm{Vol}_{\mathbb{E}^d}(B) \cdot \alpha_s^d (1 - o(1)).$$

This concludes the proof of the lemma. $\square$

**Lemma 10.** *Denote by* $A = \delta^2 \cdot \frac{(M+r+s)(M+r-s)(s+M-r)(s+r-M)}{4s^2}$. *If* $\alpha_s - \alpha_r < \alpha_M < \sqrt{\alpha_s^2 + \alpha_r^2}$, *then*

$$\Theta\left(d^{-1}\right) \leq \frac{W(\alpha_r, \alpha_M, \alpha_s) \cdot \delta}{\mathrm{Vol}_{\mathbb{E}^{d-1}}(B) \cdot A^{\frac{d+1}{2}}} \leq \Theta\left(d^{-\frac{1}{2}}\right). \tag{24}$$

*If* $\alpha_M > \sqrt{\alpha_s^2 + \alpha_r^2}$, *then*

$$\frac{\mathrm{Vol}_{\mathbb{H}^d}(B(\alpha_r)) - W(\alpha_r, \alpha_M, \alpha_s)}{\mathrm{Vol}_{\mathbb{E}^{d-1}}(B) \cdot A^{\frac{d+1}{2}} \cdot \delta^{-1}} \leq \Theta\left(d^{-\frac{1}{2}}\right). \tag{25}$$

*Proof.* The result follows directly from Lemma 7. □

**Lemma 11.** *If $M > s\sqrt{2}$, then*

$$W(\alpha_s, \alpha_M, \alpha_s) \geq \frac{V_d(R)}{n} \cdot s^{d+o(d)}.$$

*If $M \leq s\sqrt{2}$, then*

$$W(\alpha_s, \alpha_M, \alpha_s) \geq \frac{V_d(R)}{n} \cdot \left( M^2 - \frac{M^4}{4s^2} \right)^{\frac{d}{2}+o(d)}.$$

*Proof.* If $M > s\sqrt{2}$, then we can apply (22) to obtain

$$W(\alpha_s, \alpha_M, \alpha_s) \geq \Theta\left( d^{-1} \cdot (s\delta)^d \cdot \mathrm{Vol}_{\mathbb{E}^d}(B) \right) = \Theta\left( d^{-1} \cdot s^d \cdot \frac{dv_d(R)}{n} \cdot \mathrm{Vol}_{\mathbb{E}^d}(B) \right)$$

$$= \Theta\left( s^d \cdot \frac{V_d(R)}{n} \cdot \frac{\mathrm{Vol}_{\mathbb{E}^d}(B)}{\mathrm{Vol}(\mathbb{S}^{d-1})} \right) = \frac{V_d(R)}{n} s^{d+o(d)}.$$

If $M \leq s\sqrt{2}$ then we can apply (21) to obtain

$$W(\alpha_s, \alpha_M, \alpha_s) \geq \mathrm{Vol}_{\mathbb{E}^{d-1}}(B) \cdot \delta^{-1} A^{\frac{d+1}{2}} \Theta(d^{-1})$$

$$= \mathrm{Vol}_{\mathbb{E}^{d-1}}(B) \cdot \delta^{-1} \cdot \alpha_M^{d+1} \cdot \left( 1 - \frac{\alpha_M^2}{4\alpha_s^2} \right)^{\frac{d+1}{2}} \cdot \Theta(d^{-1})$$

$$= \mathrm{Vol}_{\mathbb{E}^{d-1}}(B) \cdot \delta^d \cdot M^{d+1} \left( 1 - \frac{M^2}{4s^2} \right)^{\frac{d+1}{2}} \cdot \Theta(d^{-1})$$

$$= \frac{\mathrm{Vol}_{\mathbb{E}^{d-1}}(B)}{\mathrm{Vol}(\mathbb{S}^{d-1})} \cdot \frac{V_d(R)}{n} \cdot M^{d+1} \left( 1 - \frac{M^2}{4s^2} \right)^{\frac{d+1}{2}} \cdot \Theta(1)$$

$$= \Theta\left( d^{-1} \right) \cdot \frac{V_d(R)}{n} \cdot \left( M^2 - \frac{M^4}{4s^2} \right)^{\frac{d+1}{2}} = \frac{V_d(R)}{n} \cdot \left( M^2 - \frac{M^4}{4s^2} \right)^{\frac{d}{2}+o(d)}.$$

□

**Corollary 4.** *If $s > 1$ and either $M > s\sqrt{2}$ or $M^2 - M^4/4s^2 > 1$, then there exists a constant $S > 1$ such that $W(\alpha_s, \alpha_M, \alpha_s) \geq \frac{V_d(R)}{n} S^{d+o(d)}$.*

**Lemma 12.** *Assume that $s > 1$. If*

$$M > s\sqrt{2} \text{ or } M^2 - \frac{M^4}{4s^2} > 1, \tag{26}$$

*then there exist constants $S > 1$ and $\varepsilon > 0$ such that*

$$W\left( \alpha_M, \alpha_s, \alpha_s + \varepsilon\delta \right) \geq \frac{V_d(R)}{n} S^{d+o(d)}.$$

*Proof.* If $M > s\sqrt{2}$, then we can choose a sufficiently small $\varepsilon > 0$ such that $M^2 \geq s^2 + (s+\varepsilon)^2$. Then, the result follows from Lemma 11 and the fact that $s > 1$. Otherwise $M^2 < s^2 + (s+\varepsilon)^2$, and instead of the condition $M^2 - \frac{M^4}{4s^2} > 1$ from (24) we have

$$\frac{4s^2(s+\varepsilon)^2 - (M^2 - s^2 - (s+\varepsilon)^2)^2}{4(s+\varepsilon)^2} > 1.$$

As it holds for $\varepsilon = 0$, we can choose a small enough $\varepsilon > 0$ that it is still satisfied. □

**Lemma 13.** *The dependence of consecutive steps is negligible.*

*Proof.* First, consider the case $M \leq s\sqrt{2}$. It suffices to show that $W_1 = W\left(\alpha_M, \alpha_s, \alpha_s + 2\varepsilon\delta\right)$ is negligible in comparison with $W_1 + W_2 = W\left(\alpha_M, \alpha_s, \alpha_s + \varepsilon\delta\right)$. From (24), we can write their ratio

$$\frac{W_1}{W_1 + W_2} = d^{O(1)} \cdot \left( \frac{4(s+\varepsilon)^2}{2\left(M^2 s^2 + s^2(s+\varepsilon)^2 + M^2(s+\varepsilon)^2\right) - \left(M^4 + s^4 + (s+\varepsilon)^4\right)} \right.$$
$$\left. \cdot \frac{2\left(M^2 s^2 + s^2(s+2\varepsilon)^2 + M^2(s+2\varepsilon)^2\right) - \left(M^4 + s^4 + (s+2\varepsilon)^4\right)}{4(s+2\varepsilon)^2} \right)^{\frac{d+1}{2}} = o(1).$$

Since at each step we reduce the radius of the ball centered at **q** which contains the current position by at least $\varepsilon\delta$, any ball encountered in the current iteration intersects only a constant number of other balls, so their overall effect is negligible.

In the case when $M > s\sqrt{2}$ and $s > 1$ with probability $1 - o(1)$ we find the nearest neighbor in one step. □

**Lemma 14.** *If for some constants $M$, $s$, $r$ we have $M^2 > s^2 + r^2$, then*

$$\text{Vol}_{\mathbb{H}^d}\left(B(\alpha_r)\right) - W\left(\alpha_r, \alpha_M, \alpha_s\right) \leq \text{Vol}_{\mathbb{H}^d}\left(B(\alpha_r)\right) \cdot \beta^d$$

*with some $\beta < 1$.*

*Proof.* It follows from (25) that

$$\text{Vol}_{\mathbb{H}^d}\left(B(\alpha_r)\right) - W\left(\alpha_r, \alpha_M, \alpha_s\right) \leq \Theta\left(d^{-\frac{1}{2}}\right) \cdot \text{Vol}_{\mathbb{E}^{d-1}}(B) \cdot A^{\frac{d+1}{2}} \cdot \delta^{-1}.$$

It remains to bound from above the following ratio:

$$\frac{\Theta\left(d^{-\frac{1}{2}}\right) \cdot \text{Vol}_{\mathbb{E}^{d-1}}(B) \cdot A^{\frac{d+1}{2}} \cdot \delta^{-1}}{\text{Vol}_{\mathbb{H}^d}\left(B(\alpha_r)\right)} = \frac{\Theta\left(d^{-\frac{1}{2}}\right) \cdot \text{Vol}_{\mathbb{E}^{d-1}}(B) \cdot A^{\frac{d+1}{2}}}{\text{Vol}_{\mathbb{E}^d}(B) \cdot r^d \cdot \delta^{d+1}} = \Theta(1) \cdot \left(\frac{A}{\delta^2 r^2}\right)^{\frac{d+1}{2}}$$

$$= \Theta(1) \cdot \left(\frac{(M+r+s)(M+r-s)(s+M-r)(s+r-M)}{4r^2 s^2}\right)^{\frac{d+1}{2}}$$

$$= \Theta(1) \cdot \left(1 - \frac{(M^2 - s^2 - r^2)^2}{4s^2 r^2}\right)^{\frac{d+1}{2}} \leq \beta^{d+1}$$

for some $\beta < 1$. □

## C.2 ANALYSIS OF THE BOUNDARY EFFECT

**Lemma 15.** *The boundary effect is negligible and does not affect our results.*

*Proof.* Consider the worst-case scenario when the current element is located on the boundary of the ball $\mathcal{B}$ (see Figure 6). Indeed, in this case, the largest part of the neighborhood is located outside $\mathcal{B}$, which may reduce the number of neighbors of a given element.

First, consider the Euclidean space. According to Lemma 5, this is equivalent to the hyperbolic case with $R \ll 1/\sqrt{d}$. Let us show that $W(\alpha, R, R)$ converges to $\frac{1}{2}\text{Vol}_{\mathbb{E}^d}B(\alpha)$ if $\alpha = \Theta(\delta)$ with $\delta = \left(\frac{d\text{Vol}_{\mathbb{E}^d}(B(R))}{\text{Vol}(\mathbb{S}^{d-1})n}\right)^{1/d}$. Note that $W(\alpha, R, R) = C_\alpha(h) + C_R(\alpha - h) < \frac{1}{2}\text{Vol}_{\mathbb{E}^d}B(\alpha)$. It is easy to see that $h = \alpha - \frac{\alpha^2}{2R}$. Then,

$$C_\alpha(h) = \frac{1}{2}\text{Vol}_{\mathbb{E}^{d-1}}(B)\alpha^d \text{B}\left(\frac{d+1}{2}, \frac{1}{2}; 1 - \frac{\alpha^2}{4R^2}\right)$$

$$= (1 + o(1)) \cdot \frac{1}{2}\text{Vol}_{\mathbb{E}^{d-1}}(B)\alpha^d \text{B}\left(\frac{d+1}{2}, \frac{1}{2}\right) = (1 + o(1)) \cdot \frac{1}{2}\text{Vol}_{\mathbb{E}^d}B(\alpha)$$

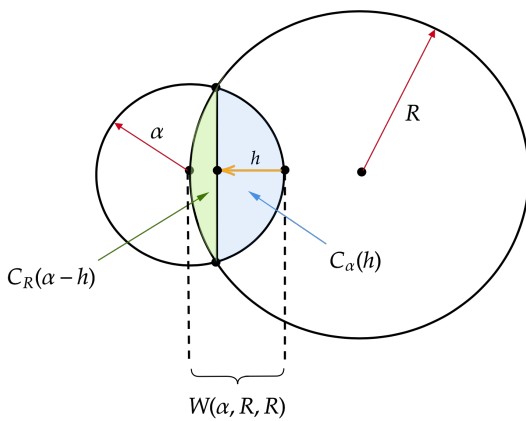

Figure 6: Illustration for the proof of Lemma 15

if $\alpha \ll 1/\sqrt{d}$. Indeed,

$$
\mathrm{B}\left(\frac{d+1}{2}, \frac{1}{2}\right) - \mathrm{B}\left(\frac{d+1}{2}, \frac{1}{2}; 1 - \frac{\alpha^2}{4R^2}\right) = \int_0^{\frac{\alpha^2}{4R^2}} t^{(d-1)/2}(1-t)^{-1/2} dt
$$

$$
\leq \frac{1}{\sqrt{1 - \frac{\alpha^2}{4R^2}}} \int_0^{\frac{\alpha^2}{4R^2}} t^{(d-1)/2} dt = \frac{2\left(\frac{\alpha}{2R}\right)^{d+1}}{(d+1)\sqrt{1 - \frac{\alpha^2}{4R^2}}} = o\left(\mathrm{B}\left(\frac{d+1}{2}, \frac{1}{2}\right)\right),
$$

since $\mathrm{B}\left(\frac{d+1}{2}, \frac{1}{2}\right) = \Theta\left(1/\sqrt{d}\right)$ and

$$
\frac{\alpha}{R} = \frac{\Theta(1)}{R} \cdot \left(\frac{d \mathrm{Vol}_{\mathbb{E}^d}(B(R))}{\mathrm{Vol}(\mathbb{S}^{d-1})n}\right)^{1/d} = \frac{\Theta(1)}{n^{1/d}} \to 0 \text{ for } d \ll \log n.
$$

In the hyperbolic case, the "small" ball of radius $\alpha$ is small enough ($\alpha \ll 1/\sqrt{d}$) so that its geometry can be considered Euclidean, and thus we can apply Lemma 5 to estimate the volume of $C_\alpha(h)$. The "large" hyperbolic ball of radius $R$, in this case, can be thought of as a Euclidean ball of radius $\sim \frac{1}{2}e^{2R}$ in the upper half–space model, and then we can carry out the same analysis on the Euclidean scale of the "small" radius $\alpha$ ball. Note that in this case

$$
\frac{\alpha}{e^{2R}} = \frac{\Theta(1)}{e^{2R}} \left(\frac{d v_d(R)}{n}\right)^{1/d} \leq \frac{\Theta(1)e^{R(d-1)/d}}{e^{2R}n^{1/d}} \to 0.
$$

$\square$

### C.3 TIME AND SPACE COMPLEXITY

In this section, we estimate time and space complexity. Recall that $N(v)$ denotes the number of neighbors of an element $v$ in $G(M)$ and $f = (n-1)\mathrm{Vol}_{\mathbb{H}^d}(B(\alpha_M))/\mathrm{Vol}_{\mathbb{H}^d}(\mathcal{B})$. The proofs below mostly follow the corresponding proofs from Prokhorenkova & Shekhovtsov (2020). We keep the proofs here for completeness.

**Lemma 16.** *The probability of the event "$N(v)$ satisfies $N(v) \leq \frac{3}{2}f$" is at least $1 - \frac{4}{f}$.*

*Proof.* Note that the random variable $N(v)$ is upper bounded by the random variable $N'(v)$ having the binomial distribution $\mathrm{Bin}\left(n-1, \frac{\mathrm{Vol}_{\mathbb{H}^d}(B(\alpha_M))}{\mathrm{Vol}_{\mathbb{H}^d}(\mathcal{B})}\right)$ with the expected value $f$. Thus, via the Chebyshev's inequality, we obtain

$$
\mathsf{P}\left(N(v) > \frac{3}{2}f\right) \leq \mathsf{P}\left(N'(v) > \frac{3}{2}f\right) \leq \mathsf{P}\left(|N'(v) - f| > \frac{f}{2}\right) \leq \frac{4\,\mathsf{Var}\left(N'(v)\right)}{f^2} \leq \frac{4}{f}.
$$

$\square$

**Lemma 17.** *The time complexity of $l$ steps of the graph-based NNS is $O\left(lfd\right)$ with probability $1 - O\left(\frac{1}{lf}\right)$.*

*Proof.* We can upper bound the number of distance computations by the random variable distributed according to $\mathrm{Bin}\left(l(n-1), \frac{\mathrm{Vol}_{\mathbb{H}^d}(B(\alpha_M))}{\mathrm{Vol}_{\mathbb{H}^d}(\mathcal{B})}\right)$ and proceed as in the proof of Lemma 16. Finally, note that one distance computation takes $\Theta(d)$. □

**Lemma 18.** *The probability of the event "$E(G)$ satisfies $E(G) \leq \frac{3}{4}fn$" is $1 - O\left(\frac{1}{fn}\right)$.*

*Proof.* We mostly follow the proof of Prokhorenkova & Shekhovtsov (2020), but some modifications are needed to handle the boundary. We use Chebyshev's inequality, so we have to estimate the variance $\mathrm{Var}E(G)$:

$$
\begin{aligned}
\mathrm{Var}E(G) &= \mathbb{E}\left(E(G)^2\right) - (\mathbb{E}E(G))^2 \\
&= \sum_{e_1, e_2 \in \binom{\mathcal{D}}{2}} \mathsf{P}(e_1, e_2 \in E(G)) - \mathsf{P}(e_1 \in E(G)) \cdot \mathsf{P}(e_2 \in E(G)).
\end{aligned}
$$

Obviously, if $e_1$ and $e_2$ do not share common endpoints, then $\mathsf{P}(e_1, e_2 \in E(G)) = \mathsf{P}(e_1 \in E(G)) \cdot \mathsf{P}(e_2 \in E(G))$. Also, one can see that if $e_1$ and $e_2$ share one element, then we also have $\mathsf{P}(e_1, e_2 \in E(G)) = \mathsf{P}(e_1 \in E(G)) \cdot \mathsf{P}(e_2 \in E(G))$. Indeed, assume that both $e_1$ and $e_2$ contain a common element $v$. Then the equality holds conditioned on the position of $v$ since both sides of the equation are defined in terms of $v$-th neighborhood size.

Thus, only the terms with $e_1 = e_2$ remain in the expression:

$$
\mathrm{Var}E(G) = \sum_{e_1 \in \binom{\mathcal{D}}{2}} \mathsf{P}(e_1 \in E(G)) - (\mathsf{P}(e_1 \in E(G)))^2 \leq \mathbb{E}E(G).
$$

Finally, we get

$$
\mathsf{P}\left(|E(G) - \mathbb{E}E(G)| > \frac{\mathbb{E}E(G)}{2}\right) \leq \frac{4\mathrm{Var}E(G)}{(\mathbb{E}E(G))^2} \leq \frac{4}{\mathbb{E}E(G)}.
$$

It remains to note that $\mathbb{E}E(G) \leq nf/2$ and due to Lemma 15 we also have $\mathbb{E}E(G) = \Theta(nf)$. From this the lemma follows.

□

## D  $k$-NN GRAPHS

Note that in Theorem 1 we consider undirected graphs since the condition on $\rho(\cdot, \cdot)$ is symmetric. In this section, we extend the theorem to nearest neighbor graphs. In this case, we assume that each node is connected to a fixed number of its nearest neighbors by directed edges. During graph traversal, outgoing edges are used to explore the neighborhood.

**Theorem 3.** *Assume that $d = d(n)$ is such that $\log \log n \ll d(n) \ll \log n$ and $R = R(n)$ is such that $e^R \ll n^{\frac{1}{d-1}} \cdot d^{-1/2}$.*

*For the exact NNS, let $M$ be a constant s.t. $M > \sqrt{2}$ and let $k = [M^d]$. Then, with probability $1 - o(1)$, NNS based on the $k$-NN graph finds the exact nearest neighbor. For $c, r$-ANN with some constant $c > 1$, let $M$ be a constant s.t. $M > \sqrt{\frac{4c^2}{3c^2-1}}$ and $k = [M^d]$. Then, with probability $1 - o(1)$, NNS based on the $k$-NN graph solves $c, r$-ANN for any $r$.*

*In both cases, the time complexity is $O\left(k \cdot n^{1/d} \cdot R \cdot (v_d(R))^{-1/d}\right)$;*

*the space complexity is $O\left(k \cdot n \cdot \log n\right)$.*

*Proof.* First, note that the space complexity is straightforward: each node has exactly $k$ outgoing edges. Similarly, the complexity of one step is $O(k \cdot d)$.

Thus, it remains to show that kNN-based NNS succeeds with probability $1 - o(1)$ and that the number of steps can be upper bounded similarly to Theorem 1. To prove this, it is sufficient to show that the distance to the $k$-th neighbor is sufficiently large, so that the convergence can be guaranteed by the proof of Theorem 1.

For every node $v$, define a random variable $\rho_k(v)$ to be the distance from $v$ to its $k$-th nearest element. It is sufficient to show that $\rho_k(v) \geq \widehat{M}\delta$, where $\delta = \left(\frac{dv_d(R)}{n}\right)^{1/d}$ and $\widehat{M}$ satisfies the required conditions on $M$ in Theorem 3. For some $\varepsilon > 0$, let $\widehat{M} := M(1 - \varepsilon)$. Obviously, by choosing small enough $\varepsilon$, we can guarantee that the required conditions are satisfied.

Note that $N_r(v)$ — the number of elements at distance at most $r$ from $v$ — can be upper bounded by a random variable having the binomial distribution $\text{Bin}\left(n - 1, \frac{\text{Vol}_{\mathbb{H}^d}(B(r))}{\text{Vol}_{\mathbb{H}^d}(\mathcal{B})}\right)$ (the upper bound is caused by possible boundary effects). If $r = \widehat{M}\delta$, then $\frac{\text{Vol}_{\mathbb{H}^d}(B(r))}{\text{Vol}_{\mathbb{H}^d}(\mathcal{B})} = \widehat{M}^d/n$. Thus, $N_{\widehat{M}\delta}(v)$ is upper bounded by $\text{Bin}\left(n, \widehat{M}^d/n\right)$.

Recall that $k = [M^d]$. It is sufficient to show that with high probability we have $N_{\widehat{M}\delta}(v) < k$ for all $v$. Using the union bound over all possible $v$ and the Chernoff's bound, we obtain the following:

$$\mathsf{P}\left(\exists v : N_{\widehat{M}\delta}(v) \geq k\right) \leq n \cdot e^{-\frac{\left(k - \widehat{M}^d\right)^2}{2\left(\widehat{M}^d + (k - \widehat{M}^d)/3\right)}} \leq e^{\log n - \frac{M^{2d}(1 - o(1))}{\Theta(M^d)}} \to 0$$

as $n \to \infty$ since $d \gg \log \log n$.

$\square$

# E  GRAPH-BASED NNS IN SPARSE REGIME

In this section, we prove Theorem 2. For this, we show that the elements are tightly concentrated near the boundary of the ball, and thus the convergence guarantees resemble those on the sphere (Prokhorenkova & Shekhovtsov, 2020).

Let us denote by $\varphi(\mathbf{x}_i, \mathbf{x}_j)$ the angle $\angle \mathbf{x}_i O \mathbf{x}_j$. For the sphere, a graph $G(M)$ is constructed by connecting such pairs of elements that (Prokhorenkova & Shekhovtsov, 2020):

$$\cos\varphi(\mathbf{x}_i, \mathbf{x}_j) \geq \sqrt{\frac{2M\log n}{d}}\,. \tag{27}$$

For the hyperbolic space, we connect the elements $\mathbf{x}_i$ and $\mathbf{x}_j$ iff

$$\cosh\rho(\mathbf{x}_i, \mathbf{x}_j) \leq \cosh^2 R - \sinh^2 R \cdot \sqrt{\frac{2M\log n}{d}}. \tag{28}$$

Note that this corresponds to the elements $\mathbf{x}_i$ and $\mathbf{x}_j$ located on the boundary of $\mathcal{B}$ with $\varphi(\mathbf{x}_i, \mathbf{x}_j)$ bounded by (27).

Recall that for $c > 1$ we set

$$\alpha_c := \frac{\cosh^2 R - \cosh(\text{arccosh}(\cosh^2 R)/c)}{\sinh^2 R}\,. \tag{29}$$

Let us explain this choice. Below we prove that NNS in the hyperbolic space behaves similarly to NNS on a sphere. In other words, we can project all the elements to the boundary of $\mathcal{B}$, and this will not significantly affect the process. However, we have to properly adjust $\alpha_c$ taking into account the multiplicative approximation factor $c$. In Prokhorenkova & Shekhovtsov (2020), $\alpha_c$ is the cosine of the angle $\frac{\pi}{2c}$. There, $\frac{\pi}{2c}$ is the spherical distance that is $c$ times smaller than the distance between the elements with $\varphi(\mathbf{x}_i, \mathbf{x}_j) = \pi/2$. The hyperbolic distance $D$ between the elements with $\varphi(\mathbf{x}_i, \mathbf{x}_j) = \pi/2$ located on the boundary of $\mathcal{B}$ satisfies the following equality:

$$\cosh D = \cosh^2 R, \text{ i.e., } D = \text{arccosh}(\cosh^2 R)\,.$$

Then, we need to compute $\cos \varphi(\mathbf{x}_i, \mathbf{x}_j)$ if $\rho(\mathbf{x}_i, \mathbf{x}_j) = D/c$. This follows from the hyperbolic law of cosines:

$$\cos \varphi(\mathbf{x}_i, \mathbf{x}_j) = \frac{\cosh^2(R) - \cosh(D/c)}{\sinh^2(R)},$$

which is equal to $\alpha_c$ in Equation (29).

Thus, it remains to show that the convergence guarantees transfer from spherical to hyperbolic space. Let us first prove that the elements are tightly concentrated near the boundary of the ball $\mathcal{B}$.

**Lemma 19.** *With probability $1 - o(1)$ all $n$ elements are located within a shell of width $\Theta\left(\frac{\log n}{d}\right)$ near the boundary of the ball $\mathcal{B}$.*

*Proof.* According to (3), we have

$$\frac{\mathrm{Vol}_{\mathbb{H}^d}(B(R - \varepsilon))}{\mathrm{Vol}_{\mathbb{H}^d}(B(R))} = \Theta\left(e^{-\varepsilon(d-1)}\right).$$

Let $\varepsilon = \frac{1}{d-1}(\log n + \varphi)$, where $\varphi = \varphi(n)$ is any function s.t. $\varphi \to \infty$ as $n \to \infty$. Then,

$$\frac{\mathrm{Vol}_{\mathbb{H}^d}(B(R - \varepsilon))}{\mathrm{Vol}_{\mathbb{H}^d}(B(R))} = \Theta\left(e^{-\log n - \varphi}\right) = o(1/n).$$

From this the lemma follows. $\square$

**Lemma 20.** *If two elements $\boldsymbol{x}_i, \boldsymbol{x}_j$ satisfy (27) with constant $M + \varepsilon$ for some $\varepsilon > 0$, then with probability $1 - o(1)$ they satisfy (28) with constant $M$.*

*Proof.* Let the distances from $\mathbf{x}_i$ and $\mathbf{x}_j$ to the center of the ball $\mathcal{B}$ be $r_i$ and $r_j$, respectively. From Lemma 19, $r_i$ and $r_j$ are equal to $R - \Theta\left(\frac{\log n}{d}\right)$ with high probability. Using the hyperbolic law of cosines, we get

$$\cosh \rho(\mathbf{x}_i, \mathbf{x}_j) = \cosh r_i \cosh r_j - \sinh r_i \sinh r_j \cos \varphi(\mathbf{x}_i, \mathbf{x}_j)$$

$$\leq \cosh r_i \cosh r_j - \sinh r_i \sinh r_j \sqrt{\frac{2(M + \varepsilon) \log n}{d}}$$

$$= \cosh^2(R - \Theta(\log n/d)) - \sinh^2(R - \Theta(\log n/d)) \sqrt{\frac{2(M + \varepsilon) \log n}{d}}.$$

It is sufficient to show that

$$\cosh^2(R - \Theta(\log n/d)) - \sinh^2(R - \Theta(\log n/d)) \sqrt{\frac{2(M + \varepsilon) \log n}{d}}$$

$$\leq \cosh^2 R - \sinh^2 R \cdot \sqrt{\frac{2M \log n}{d}}.$$

Note that $\cosh(R - \Theta(\log n/d)) = \frac{e^R}{2}(1 + \Theta(\log n/d))$. The same holds for $\sinh(R - \Theta(\log n/d))$. Therefore, we need

$$\frac{e^{2R}}{4}(1 + \Theta(\log n/d))\left(1 - \sqrt{\frac{2(M + \varepsilon) \log n}{d}}\right) \leq \frac{e^{2R}}{4}(1 + \Theta(e^{-2R}))\left(1 - \sqrt{\frac{2M \log n}{d}}\right).$$

Now we divide the right hand side by the left hand side and obtain

$$\frac{\left(1 - \sqrt{\frac{2M \log n}{d}}\right)}{\left(1 - \sqrt{\frac{2(M+\varepsilon) \log n}{d}}\right)}\left(1 + \Theta(\log n/d) + \Theta\left(e^{-2R}\right)\right)$$

$$= \left(1 + \left(\sqrt{\frac{2(M + \varepsilon) \log n}{d}} - \sqrt{\frac{2M \log n}{d}}\right)\left(1 + \Theta\left(\sqrt{\log n/d}\right)\right)\right)\left(1 + \Theta(\log n/d) + \Theta\left(e^{-2R}\right)\right)$$

$$= \left(1 + \Omega(\sqrt{\log n/d})\right)\left(1 + \Theta(\log n/d) + \Theta\left(e^{-2R}\right)\right).$$

From this the lemma follows, since $\sqrt{\log n/d} \gg \log n/d$ and $\sqrt{\log n/d} \gg e^{-2R}$. $\square$

Let us choose such small $\varepsilon > 0$ that the required condition on $M$ is also satisfied for $M + \varepsilon$. It follows from Lemma 20 that our graph $G(M)$ contains the spherical nearest neighbor graph defined via (27) with constant $M + \varepsilon$. Thus, we may follow the proof of Theorem 2 in (Prokhorenkova & Shekhovtsov, 2020) (Appendix B.5) with the corresponding nearest neighbor graph.

Note that all spherical caps and distances mentioned in the proof of (Prokhorenkova & Shekhovtsov, 2020) are defined by conditions $\cos \varphi \geq \sqrt{\frac{2S \log n}{d}}$ for some constant $S$. According to Lemma 20, such constants are negligibly affected by deviations from the boundary, and thus we can directly follow the proof of Prokhorenkova & Shekhovtsov (2020) to obtain the convergence guarantees and complexity. Indeed, all requirements for the constants in the proof are strict, and thus a sufficiently small $\varepsilon$ can be chosen such that they are still satisfied.

## F  Experiments

**Best-first search**  In the main text, for the synthetic experiments, we use the greedy search. However, in graph-based NNS algorithms, the greedy search is often replaced by the best-first search (a.k.a. beam search). In this case, instead of keeping track of only the best candidate, we maintain a dynamic list of several candidates. In particular, this technique is used in HNSW (Malkov & Yashunin, 2018), and the size of the candidate list is controlled by the parameter $efSearch$.

**Hyperparameters**  Recall that to plot Figure 1, we vary the graph degree to control the complexity and accuracy of NNS. In Figures 2 and 3, we use the best-first search, so we vary the parameter $efSearch$ and fix the graph degree. In this case, we tune $M$ and $efConstruction$ HNSW hyperparameters, which are responsible for the graph degree and quality of the graph construction, respectively. For both GloVe and Poincaré GloVe databases, we get $M$=10 and $efConstruction$=300. For other datasets, we get $M$=8.

**NSG**  In addition to HNSW, we also experiment with the NSG graph (Fu et al., 2019). For this algorithm, the degree is controlled by the parameter $R$, and the optimal degree is $R = 2M$.

**Spherical Shell implementation**  As a baseline, we implement Spherical Shell proposed by Wu & Charikar (2020). As in the original paper, we split the elements into 25 bands such that $w^{i-1} \leq \frac{1}{1-\|x\|^2} \leq w^i$. To achieve this, we take $w = 1.05$. For each band, we construct LSH with 50 tables. Each table has $\lceil \log(m) \rceil - 1$ bins, where $m$ is the number of elements in the band. The number of probes is tuned for each band to reach 0.95 recall (this value has been chosen to achieve better results). When searching for the nearest neighbor for a query $q$, we compute the nearest neighbors within $l$ bands closest to $q$ and take the best one according to the hyperbolic distance. The number of bands $l$ is varied from 1 to 10, thus giving the curve on Figure 2.

**Additional datasets**  In addition to the standard Poincaré GloVe dataset, we also consider the Poincaré GloVe embeddings trained for $d = 10$ using the source code provided by the authors (Tifrea et al., 2019). Additionally, we also take the recommendation dataset from Shevkunov & Prokhorenkova (2021), where a DSSM model was trained on a Wikipedia search dataset of (search query, relevant page) pairs. We use their Euclidean and hyperbolic representations for $d = 10$.

**Additional experiments on real datasets**  Figure 7 compares graph-based methods with Spherical Shell on the standard Poincaré GloVe. Compared to the main text, here we add the NSG algorithm. While HNSW outperforms NSG, both are better than Spherical Shell.

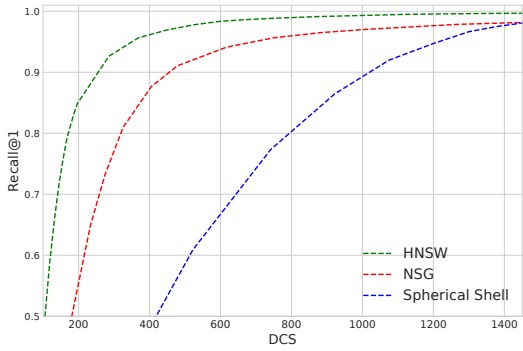

Figure 7: Graph-based methods vs Spherical Shell on Poincaré GloVe embeddings

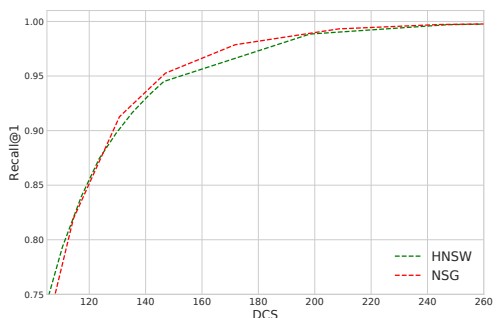

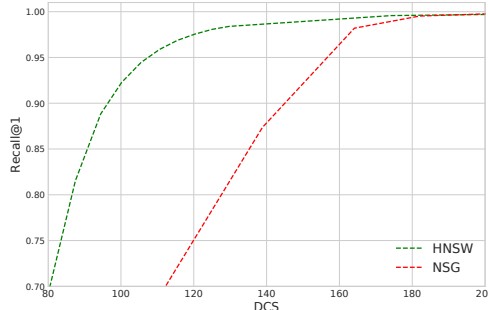

Figure 8: Comparison of algorithms on Poincaré GloVe with $d = 10$

Figure 9: Comparison of algorithms on the recommendation dataset with $d = 10$

Figures 8 and 9 compare the algorithms on the Poincaré GloVe dataset with $d = 10$ and the recommendation dataset with $d = 10$ from Shevkunov & Prokhorenkova (2021). We see that HNSW and NSG have similar performance on Poincaré GloVe, while HNSW outperforms NSG on the recommendation dataset. In contrast, Spherical Shell needs a significantly larger number of distance computations, so it is not shown in the figures. On the recommendation dataset, low performance of Spherical Shell can be explained by the fact that this data significantly differs from the Poincaré GloVe: the learned vector norms are much larger. Because of this, we either have annuli of large width (significant approximation error) or many annuli (increased total complexity). Also, it is important to note that Spherical Shell is well suited for sparse datasets, while we have $d = 10$ in this experiment.

Then, we compare the efficiency of NNS in Euclidean and hyperbolic spaces. For this, we consider the embeddings of the two datasets discussed above into Euclidean and hyperbolic spaces and perform NNS based on HNSW for the same set of queries. Figure 10 shows that hyperbolic NNS is more efficient.

**Additional synthetic experiments** Figure 11 is similar to Figure 1, but now we use the best-first search instead of the greedy search. We see that the conclusions are the same.

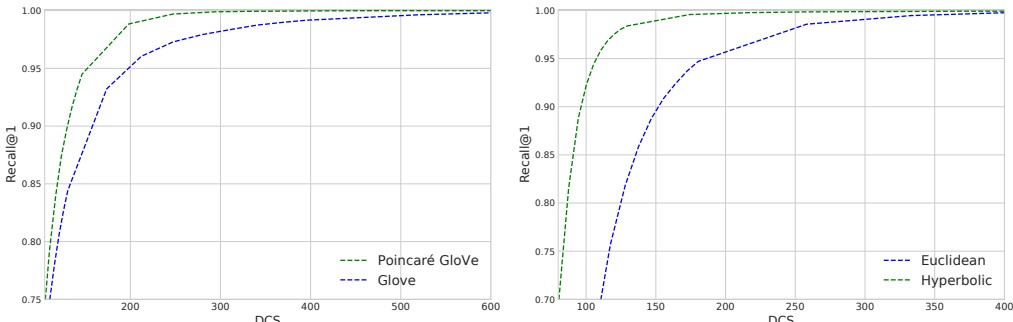

Figure 10: Efficiency of HNSW in hyperbolic space vs Euclidean space on GloVe with $d = 10$ (left) and recommendations (right)

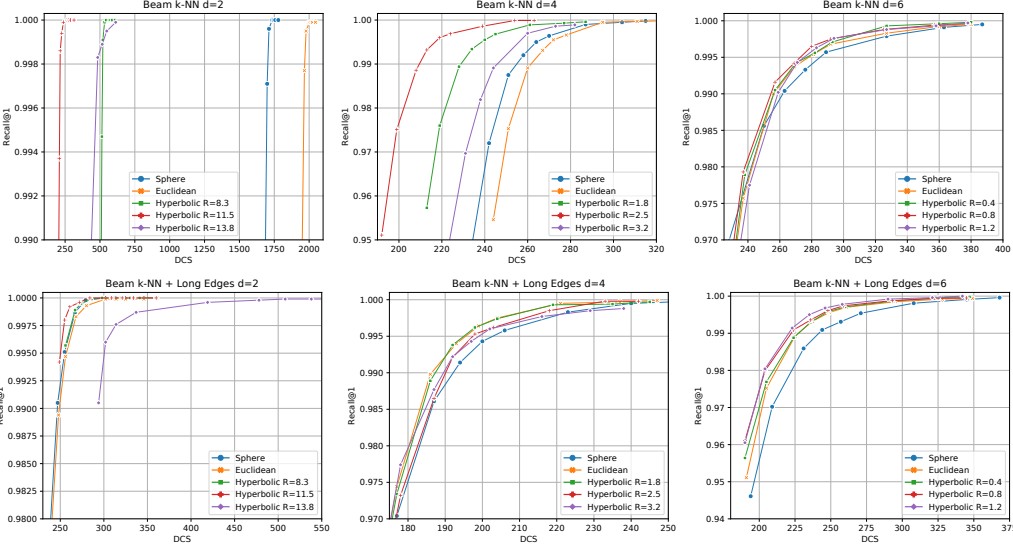

Figure 11: Best-first search over NN graphs without (top row) and with (bottom row) long edges

