# OpenReview forum: "Graph-based Nearest Neighbor Search in Hyperbolic Spaces"
_ICLR.cc/2022/Conference — ICLR 2022 Poster_

### Official Review · Reviewer_VNYQ · 2021-11-02

**Correctness:** 4
**Technical Novelty And Significance:** 3
**Empirical Novelty And Significance:** 3
**Recommendation:** 6
**Confidence:** 4

**Main Review:**

Thanks for the interesting paper. It is interesting to know that the graph-based method, which works very well for the Euclidean space, also works for the hyperbolic space. The analysis is solid and the experiments justify the theoretical findings.

=================

I have read the reviews of other reviewers and the author response. I still think that the paper is of value by extending the analysis of Euclidean space (an ICML’21) to hyperbolic space. However, I agree the experiments should be improved to include more datasets/graph-based similarity search algorithms.

**Summary Of The Paper:**

This paper proposes to use the graph-based method for similarity search in hyperbolic space. Extensive analysis shows that graph-based method achieves sublinear time complexity for hyperbolic space and its performance is even better than for the Euclidean space under some mild conditions. Empirical experiments are also conducted to validate the theoretical analysis.

**Summary Of The Review:**

I think this is a solid and interesting paper.

---

> ### Author Response · Authors · 2021-11-11
> **Reply to Reviewer VNYQ**
>
> Thank you very much for your positive feedback!

---

> ### Author Response · Authors · 2021-11-22
> **Revised paper**
>
> We’ve updated the paper. We added more datasets and an additional graph-based NNS algorithm (Sections 5.1 and Appendix F). We also added the theoretical analysis of the sparse regime (Section 4.3 and Appendix E). We hope this resolves your concerns. If you have any additional comments or suggestions, we will be happy to address them!

---

> ### Author Response · Authors · 2021-12-02
> **Revised paper**
>
> Dear Reviewer, We would like to ask whether your additional concerns raised during the discussion period have been addressed in the revised version of the paper. Your feedback and comments would help us to improve the paper. Thank you!

---

### Official Review · Reviewer_6ftD · 2021-11-07

**Correctness:** 3
**Technical Novelty And Significance:** 3
**Empirical Novelty And Significance:** Not applicable
**Recommendation:** 6
**Confidence:** 3

**Main Review:**

This paper analyzed best-first-search graph-based NNS algorithms in the hyperbolic space theoretically under some assumptions and practically to the Poincar{/'e} Glove word embeddings, which is an interesting work to attract research of NNS in hyperbolic space.

Theoretically the authors do show that under some assumptions and in some cases, the graph-based NNS in hyperbolic space can be more efficient than that in the Euclidean space, though clearly these assumptions probably not hold in realistic dataset. Further the authors apply the methodology to the Poincar{/'e} Glove data, and the experiments help justify the efficiency of search over hyperbolic space.

However, there are some weakness:
1. the assumptions are strong, it looks like to be common in this area to assume that data elements are distributed uniformly within a sphere in the space? also the dense setting, where one must either have a lot of data points or use a low dimensional hyperbolic space.
2. the experiments over Poincar{/'e} Glove word embeddings look good, however, is it possible to include some other datasets and varies the dimension in a realistic setting (not only for the synthetic  datasets)? which I believe would help convince readers of the efficiency of the algorithms for real datasets.

Also I have some questions regarding the paper
In the paper, the authors use the assumption that data uniformed distributed in the ball of radius R in the hyperbolic space, what's the difference to uniform over the sphere? Particularly in high dimension case, uniform over the ball would be close to uniform over sphere (at least in Euclidean space), what results would be caused, can the authors explain more on this?


**Summary Of The Paper:**

This paper looks into graph-based algorithms for nearest neighbor search (NNS) problem in the hyperbolic space.

Theoretically, the authors derived the time and space complexity of graph-based NNS under some assumptions (data uniformed distributed in the ball, low dimensional dense), and in some cases, graph-based NNS has lower time complexity in the hyperbolic space.

Practically, the authors apply the methodology to Poincar{/'e} Glove word embeddings, where it outperforms some existing baselines.

**Summary Of The Review:**

This paper is an interesting work to analyze best-first-search graph-based NNS algorithms in the hyperbolic space.

---

> ### Author Response · Authors · 2021-11-11
> **Reply to Reviewer 6ftD**
>
> Thank you very much for your review! Let us address the concerns.
>
> > 1. the assumptions are strong, it looks like to be common in this area to assume that data elements are distributed uniformly within a sphere in the space? also the dense setting, where one must either have a lot of data points or use a low dimensional hyperbolic space.
>
> The uniformity assumption is indeed common in this area. Existing theoretical results for the Euclidean space are based on this assumption (Laarhoven, 2018; Prokhorenkova & Shekhovtsov, 2020). Thus, to compare our results with the existing literature, we use a similar setup. However, we hope that our analysis for the hyperbolic space and the results of Prokhorenkova & Shekhovtsov (2020) for the Euclidean one will further be extended to more complex settings.
>
> Regarding the restriction on dimensionality, this concern is addressed in the third paragraph of Section 4.2, and related issues were also discussed before in the literature,  e.g., in Prokhorenkova & Shekhovtsov (2020). We argue that for uniform datasets, small values of $d$ are close to reality. The reasons are the following:
> - While in practice datasets are often embedded into spaces of large dimension, they usually have much smaller intrinsic dimension. Informally speaking, a dataset can be locally approximated by a close-to-uniform distribution in a lower-dimensional space, and thus our theoretical results can be applied.
> - Appendix F.1 of Prokhorenkova & Shekhovtsov (2020) provides additional arguments on the dense regime being reasonable.
> - Also, according to Prokhorenkova & Shekhovtsov (2020), in the sparse regime, graph-based NNS is expected to converge in just two steps (for the uniform distribution). This contradicts practical observations, thus implying that the dense setup is more realistic.
> - Similarly, the optimal graph degree observed in practice better agrees with the theoretical guarantees in the dense regime.
>
> > 2. the experiments over Poincar{/'e} Glove word embeddings look good, however, is it possible to include some other datasets and varies the dimension in a realistic setting (not only for the synthetic  datasets)?
>
> To illustrate the generalizability of our conclusions, we plan to add experiments on a dataset from another domain (recommendations). We will update the paper when we get the results.
>
> > In the paper, the authors use the assumption that data uniformed distributed in the ball of radius R in the hyperbolic space, what's the difference to uniform over the sphere? Particularly in high dimension case, uniform over the ball would be close to uniform over sphere (at least in Euclidean space), what results would be caused, can the authors explain more on this?
>
> It is a good and non-trivial question. For now, we do not have a formal explanation for what will happen for very large values of $d$. Indeed, when $d \gg \log n$, one can show that all elements are located within a spherical shell of a width of order $\frac{\log n}{d} \ll 1$ near the boundary of the ball. However, these small differences in the radii can be significant and can change relative distances between the elements, since the angular distances between all pairs of elements are close to $\pi/2$.

---

> ### Author Response · Authors · 2021-11-22
> **Revised paper**
>
> We’ve updated the paper. In particular, we added more datasets and an additional graph-based NNS algorithm (Sections 5.1 and Appendix F).
>
> We would like to thank you for your suggestion to consider the analogy between the sparse regime and the spherical case. We managed to formalize this and added the analysis of the sparse regime to the paper, see Section 4.3 and Appendix E.
>
> If you have any additional comments or concerns, we will be happy to address them!

---

### Official Review · Reviewer_zLTN · 2021-11-10

**Correctness:** 3
**Technical Novelty And Significance:** 3
**Empirical Novelty And Significance:** 1
**Recommendation:** 6
**Confidence:** 4

**Main Review:**

Improving NNS for hyperbolic embeddings is potentially of significant practical importance as previous attempts to build production systems based on this geometry (e.g. recommender systems) has been hampered by the lack of efficient recall systems in the very large n domain. As this is the area the authors focus on, this is potentially a significant work in the development of machine learning methods that utilise hyperbolic geometry. The paper is also very nicely written, being both clear and without errors.

The main result of the paper is that under certain assumptions, NNS in hyperbolic space can be faster than in Euclidean space.

My main concerns are around the validity of the assumptions and the empirical evaluation.

The assumption that d << log n seems very reasonable, but the data elements being uniformly distributed within a ball of radius R seems much more restrictive, as the authors point out. While previous works also assume uniform data distributions, I would like the authors to comment to what extent their theoretical results would generalise to more realistic settings. While I agree with the statement that points that are uniformly distributed in the Poincare ball can be used to generate complex networks, in practice, the NNS would not function on a set of points (presumably generated by some ML system) that have this property and I am interested in how the performance guarantees change as this restriction is relaxed.

Can the authors please also comment on this statement "The obtained time and space complexity heavily relies on this radius: for small values, the obtained results are similar to the spherical case, while for larger values, the exponential growth of hyperbolic volumes (see the next section) allows us to obtain even faster convergence." Which domain is likely to be encountered in practice? I would have thought that R was a design decision affected by e.g. choices on learning rate or initialisation in the ML system producing the hyperbolic representations.

Having established theoretical results for d << log n, the empirical evaluation is somewhat disappointing. Why did the authors choose to do the theoretical analysis for the dense case and then empirically analyse the sparse case? Open source code exists that would allow the authors to generate hyperbolic embeddings of a lower dimension. Could the authors perform this experiment?

In Section 5.1 and 5.2, the outperformance of HNSW is not particularly surprising given that this method was itself inspired by hyperbolic geometry and it's unclear if this is simply an artefact of embedding a tree structured dataset. Could the authors try additional datasets, for instance Wordnet at least?


The following citation is also missing "Neural Embeddings of Graphs in Hyperbolic Space", Chamberlain, Clough and Deisenroth, 2017.

**Summary Of The Paper:**

This paper addresses the important problem of nearest neighbour search for data that lives in hyperbolic space and provides a novel result on the relative time complexity of NNS in hyperbolic and Euclidean spaces under some assumptions.

**Summary Of The Review:**

Interesting and potentially impactful work, but I would like to see more comment on the validity of the assumptions and improved empirical evaluation before raising my score.

---

> ### Author Response · Authors · 2021-11-11
> **Reply to Reviewer zLTN - part 1**
>
> Thank you very much for the thorough review and valuable suggestions. Let us address the concerns.
>
> > While previous works also assume uniform data distributions, I would like the authors to comment to what extent their theoretical results would generalise to more realistic settings. … in practice, the NNS would not function on a set of points (presumably generated by some ML system) that have this property and I am interested in how the performance guarantees change as this restriction is relaxed.
>
> We agree that the uniformity assumption is strong, but it allows us to obtain rigorous theoretical results and directly compare them to the previous research. Regarding generalization to more complex scenarios, our work generalizes to cases when the density is not constant but is regular in some sense. For instance, if for some density $\rho$ our theorem holds, then it also holds for any density with $\Theta(\rho)$ behavior, i.e., if the density varies between $c_1 \rho$ and $c_2 \rho$ for come $c_1, c_2 > 0$. Some more general cases are also possible, but the formal proof would require much more technical work. Informally speaking, the conditions on $d$, $R$, and $n$ define the density of elements within the hyperbolic space. If the required conditions are locally satisfied, then the convergence with probability $1-o(1)$ holds for NNS based on kNN graphs with $k=M^d$. In practice, our results can be applied to real datasets, and the theoretical guarantees depend on the so-called intrinsic dimension. If we say that the intrinsic dimension is $d’$, then, informally speaking, a dataset can be locally approximated by a close-to-uniform distribution in a $d’$-dimensional space, and then our convergence results can be applied. We will add a discussion about this to the paper. While formalization is technically difficult, this intuition indeed helps to understand the results better; thank you for the comment.
>
> > Can the authors please also comment on this statement "The obtained time and space complexity heavily relies on this radius: for small values, the obtained results are similar to the spherical case, while for larger values, the exponential growth of hyperbolic volumes (see the next section) allows us to obtain even faster convergence." Which domain is likely to be encountered in practice? I would have thought that R was a design decision affected by e.g. choices on learning rate or initialisation in the ML system producing the hyperbolic representations.
>
> In practice, $R$ corresponds to how large the learned distances between the elements are, i.e., how much space the learned embedding occupies. This is indeed defined by the learning process, i.e., the distances are chosen to optimize the desired embedding loss function. For datasets that naturally fit the hyperbolic space (e.g., those with a hierarchical structure), we do not expect $R$ to be too small: small $R$ values correspond to Euclidean geometry, but we want to use hyperbolic spaces instead. Thus, if our embedding utilizes the hyperbolic exponential volume growth, then we also get faster convergence.
>
> Let us also comment that $R$ is directly related to the sectional curvature $c$, so in our analysis w.l.o.g. we assume $c=1$ and vary $R$. In practice, $c=1$ is also often assumed, but in some cases, $c$ is optimized or set based on hyperbolicity of the dataset, e.g., see Mirvakhabova et al. (2020). Similar to the intuition above, $c$ is larger for datasets that are "more hyperbolic".
>
> Mirvakhabova L. et al. Performance of hyperbolic geometry models on top-N recommendation tasks. RecSys, 2020.

---

> > ### Comment · Reviewer_zLTN · 2021-11-24
> > **Definition of the intrinsic dimension**
> >
> > Thanks for the reply. I agree that the interesting case has large R where your method delivers benefits and I am satisfied now with that aspect of the work. Please could you provide more details on the intrinsic dimension. How are you defining it and how can it be determined? I couldn't find this in the paper.

---

> > > ### Author Response · Authors · 2021-11-25
> > > **On intrinsic dimension**
> > >
> > > There are several ways to define and estimate the intrinsic dimension for a given dataset. Usually, estimation methods are based on the following idea. Consider a uniform distribution in a $d$-dimensional space. Then, we can consider some characteristics of this data depending on $d$. For a given dataset, we can measure the values of these characteristics and see how well they agree with what we expect for the $d$-dimensional uniform case.
> > >
> > > For instance, one known approach is to compute the distances to $k$-th nearest neighbors. Then, we can compare these values with what we expect for $d$-dimensional data. We can use the maximum likelihood estimate to find the best dimension (Levina and Bickel, 2005). This approach is used, e.g., in Lin & Zhao (2019) and recently in Pope et al. (2021) for images.
> > >
> > > A similar approach is to consider a so-called expansion constant $c$. This constant reflects how fast the volume of a ball grows when we double its radius. Here for discrete data, we say that the volume of a ball of radius $r$ is the number of elements within this distance from a given element. The constant $c$ is related to intrinsic dimension as $c \sim 2^d$.
> > >
> > > Another approach is to consider a doubling constant, which is the minimum value $c$ such that every ball in the dataset can be covered by $c$ balls of half the radius. For the last two methods, we refer to, e.g., Beygelzimer et al. (2006) and references therein.
> > >
> > > The intrinsic dimension is also related to a so-called disorder constant, see Goyal et al. (2008).
> > >
> > > We will add the discussion on intrinsic dimension to the appendix. Thank you for the comment!
> > >
> > > E. Levina and P. J. Bickel. Maximum likelihood estimation of intrinsic dimension. Advances in Neural Information Processing Systems, 2005.
> > >
> > > P. Pope et al. The intrinsic dimension of images and its impact on learning. International Conference on Learning Representations, 2021.
> > >
> > > N. Goyal, Y. Lifshits, H. Schütze. Disorder inequality: a combinatorial approach to nearest neighbor search. Proceedings of The International Conference on Web Search and Data Mining, 2008.

---

> ### Author Response · Authors · 2021-11-11
> **Reply to Reviewer zLTN - part 2**
>
> > Having established theoretical results for d << log n, the empirical evaluation is somewhat disappointing. Why did the authors choose to do the theoretical analysis for the dense case and then empirically analyse the sparse case? Open source code exists that would allow the authors to generate hyperbolic embeddings of a lower dimension. Could the authors perform this experiment?
>
> Note that our empirical analysis on synthetic data is performed for the dense case to correspond to our theoretical analysis. For the GloVe dataset, we deliberately take "realistic" data. In other words, we want to illustrate a more general idea that graph-based NNS works in different scenarios, not only those covered by our theoretical analysis. The Poincare GloVe embeddings for $d = 100$ are known to outperform Euclidean ones for some tasks, so it is a realistic example of a practical task. Also, the fact that we have $d = 100$ does not mean that the intrinsic dimension is large, while theoretical results depend on the intrinsic dimension. However, we agree that trying smaller values of $d$ is also interesting, and we are planning to perform similar experiments for smaller $d$. We will add the results to the revised version of the paper when we get them.
>
> > In Section 5.1 and 5.2, the outperformance of HNSW is not particularly surprising given that this method was itself inspired by hyperbolic geometry and it's unclear if this is simply an artefact of embedding a tree structured dataset. Could the authors try additional datasets, for instance Wordnet at least?
>
> To illustrate the generalizability of our conclusions, we plan to add experiments on a dataset from another domain (recommendations). We will update the paper when we get the results.
>
> > The following citation is also missing "Neural Embeddings of Graphs in Hyperbolic Space", Chamberlain, Clough and Deisenroth, 2017.
>
> Thanks, we’ll add it to the bibliography.

---

> > ### Comment · Reviewer_zLTN · 2021-11-24
> > **Nice results, thanks**
> >
> > Thank you for adding additional results. I believe these strengthen the paper as the low d case is often considered to be the regime where hyperbolic embeddings particularly shine (see e.g. Nickel and Kiela). One small thing, it would be best to avoid red-green in figure 8 due to the prevalence of people who can't distinguish these colours.

---

> ### Author Response · Authors · 2021-11-22
> **Revised paper**
>
> We’ve updated the paper. We added more datasets and an additional graph-based NNS algorithm (Sections 5.1 and Appendix F). In particular, we now have hyperbolic embeddings for a lower dimension. We also added the theoretical analysis of the sparse regime (Section 4.3 and Appendix E) and updated the text by adding a discussion on how our theoretical results relate to practice.
>
> If you have any additional comments or concerns, we will be happy to address them!

---

> > ### Comment · Reviewer_zLTN · 2021-11-26
> > **Updated review**
> >
> > Thank you for providing thorough answers to my questions and for the additions to the manuscript. I have raised my score accordingly.

---

### Official Review · Reviewer_XgPX · 2021-11-11

**Correctness:** 4
**Technical Novelty And Significance:** 3
**Empirical Novelty And Significance:** 3
**Recommendation:** 6
**Confidence:** 3

**Main Review:**

Strong points:
1.	The paper is clearly written and has the main ingredients of theoretical analysis accompanied with experiments and also experiments that attempt to validate the theoretical results.
2.	I find the insight about graph based methods in hyperbolic spaces to be interesting, even though I find the bounds on parameters quite restricting, in particular since the problem of nearest neighbor search aims at high dimensions and the authors demonstrate their work in a dimension up to 6.

Weak points:
1.	Im less concerned with the uniformity assumption, though the lack of uniformity can also happen on the low dimensional manifold even just because of sampling rates in realistic scenarios. What bothers me more is the use of the low dimension in section 5.3. if the ambient dimension is so small what is the challenge in the NNS?
2.	If the authors need to address the intrinsic dimension, I recommend a separate parameter d’
3.	More insight for practitioners could be provided on the algorithmic setting: any parameters in the graph based method that should be tuned according to this constraints and insights? E,g,: The algorithmic role of R should be explained (e.g. in the c,r-ANN).
4.	Some more motivation on the advantage of hyperbolic spaces for real life settings can enhance the view on the impact of the method.
5.	Use of M is rather confusing, in (2) it is a Riemannian manifold but later it is a scalar.


**Summary Of The Paper:**

The paper approaches the problem of nearest neighbor search in hyperbolic spaces with focus on graph methods. The authors claim to obtain an acceleration in hyperbolic spaces over Euclidean ones using graph based NNS method (e.g. Prokhorenkova 2020). The provide bounds on the dimension and Radius of search in the dense  NN domain (ie d<<logn). Experimental validation is presented which demonstrate the advantage of graph methods and also tradeoffs on efficiency in different parameters regimes to validate the theoretical analysis.

**Summary Of The Review:**

Overall the insights of the paper are important but their validity in the practical domain should be better explained in the paper.

---

> ### Author Response · Authors · 2021-11-12
> **Reply to Reviewer XgPX**
>
> Thank you for the valuable feedback! Let us address the concerns.
>
> > I find the bounds on parameters quite restricting, in particular since the problem of nearest neighbor search aims at high dimensions and the authors demonstrate their work in a dimension up to 6.
>
> Let us stress that the dimension of uniformly distributed data is considerably different from the ambient dimension of real data. Thus, even when the dataset is embedded in a high-dimensional space, it usually has a much smaller local intrinsic dimension. Informally speaking, a dataset can be locally approximated by a close-to-uniform distribution in a lower-dimensional space, and thus our theoretical results can be readily applied.
>
> > What bothers me more is the use of the low dimension in section 5.3. if the ambient dimension is so small what is the challenge in the NNS?
>
> Section 5.3 aims to illustrate the obtained theoretical results, while Sections 5.1-5.2 are about practical scenarios. To address this comment, we plan to:
> - Add a more detailed discussion regarding intrinsic dimension and connections between theory and practical datasets
> - Add experiments with larger dimensions in Section 5.3
> - Add a discussion about what is expected theoretically for sparse uniform hyperbolic datasets
>
> > If the authors need to address the intrinsic dimension, I recommend a separate parameter d’
>
> Thank you, we'll use this.
>
> > More insight for practitioners could be provided on the algorithmic setting: any parameters in the graph based method that should be tuned according to this constraints and insights? E,g,: The algorithmic role of R should be explained (e.g. in the c,r-ANN).
>
> The main aim of our theoretical analysis is to provide intuition about why classic graph-based NNS is supposed to work well not only in Euclidean space but also in the hyperbolic one.
>
> Regarding the role of R, in practice, it corresponds to how large the learned distances between the elements are, i.e., how much space the learned embedding occupies. These distances are usually defined by the learning process, i.e., they are chosen to optimize the desired embedding loss function. Note that R is usually not too small (since, in this case, the embedding is essentially Euclidean) and not too large (large values of R cause computational issues). Our experiments in Sections 5.1-5.2 show that for naturally obtained embeddings, classic graph-based algorithms work well.
>
> Thus, our suggestion for practitioners is to apply standard graph-based NNS (adapted for hyperbolic distances) and perform standard parameter tuning for them.
>
> > Some more motivation on the advantage of hyperbolic spaces for real life settings can enhance the view on the impact of the method.
>
> Thank you, we will address this comment in the revised version by adding more references and motivation.
>
> > Use of M is rather confusing, in (2) it is a Riemannian manifold but later it is a scalar.
>
> Thanks for noticing; we’ll correct this.

---

> ### Author Response · Authors · 2021-11-22
> **Revised paper**
>
> We’ve updated the paper. In particular, we added more datasets and an additional graph-based NNS algorithm (Sections 5.1 and Appendix F). We also added the theoretical analysis of the sparse regime, where we show that the dataset is essentially spherical in this case (Section 4.3 and Appendix E). Finally, we updated the text by adding more motivation and discussion on how our theoretical results relate to practice.
>
> If you have any additional comments or concerns, we will be happy to address them!

---

### Official Review · Reviewer_1ri4 · 2021-11-12

**Correctness:** 4
**Technical Novelty And Significance:** 2
**Empirical Novelty And Significance:** 2
**Recommendation:** 5
**Confidence:** 4

**Main Review:**

### Strengths

1. The paper's primary strength is the result given by Theorem 1, which rigorously analyzes the time and space complexity of a graph-based NNS for datasets that are both uniform and dense in a ball contained in hyperbolic space. It is worth noting that this result is a direct hyperbolic analog of Theorem 1 in Prokhorenkova & Shekhovtsov (2020).

### Weaknesses

1. Although the paper extends the result of Prokhorenkova & Shekhovtsov (2020) for dense graph-based NNS in Euclidean space to hyperbolic space, it does not attempt to provide theory for the more interesting and practical sparse setting. This makes the theory appear incomplete, relative to existing results in Euclidean space.

2. A large portion of the paper lacks novelty. The structure of the paper, up to and including the labelling of results/sections in the appendix, closely resembles Prokhorenkova & Shekhovtsov (2020). Certainly, the paper did have to introduce some lemmas, e.g. Lemma 1 followed from a fairly straightforward proof that allows one to bound volumes of hyperbolic balls by volumes of Euclidean ball intersections for small radii. Otherwise, a lot of the Euclidean analysis is taken straight from Appendix A of Prokhorenkova & Shekhovtsov (2020), and the hyperbolic constructions are de facto straightforward analytic extensions. Given the close resemblance of the paper to Prokhorenkova & Shekhovtsov (2020), there is an especially large amount of pressure on the paper to do a good job on being thorough with considered settings, but this was not provided: as mentioned above, no theoretical results for the sparse setting are given (in contrast to Prokhorenkova & Shekhovtsov (2020)).

3. I found the empirical results to be lacking considerably. First of all, the uniformity assumption and the assumption of small dimension was not addressed in the application section. Prokhorenkova & Shekhovtsov (2020) address these assumptions directly, both by employing dimensionality reduction and by making use of a technique that makes datasets more uniform while approximately preserving distances (Sablayrolles et al. (2018)). The argument in Section 5.2 seems to be that showing reasonable performance in practice without explicitly addressing these assumptions is sufficient, but this seems at best handwavy. These assumptions were critical for proper theoretical analysis, and one should either provide a way to bridge the gap between theory and practice, or provide a theoretical result about degradation when the assumptions do not hold.

4. The baseline seems too weak/non-exhaustive for Section 5.1 (Figure 1). The comparison is given only with respect to a single method from Wu & Charikar (2020). I understand that Spherical Shell may be the most "practical" method, but having a more proper comparison against the two other methods introduced in that paper is warranted, given the lack of prior work/baselines. Having a single baseline on 1 dataset in the sparse setting does not give me a convincing impression of the method's performance.

5. Along the same vein, I am left unconvinced by the results in Section 5.2 (Figure 2) that switching to hyperbolic embeddings from Euclidean embeddings makes NNS more efficient, or even that efficiency is not reduced. Although Figure 2 shows that HNSW built on embeddings in hyperbolic space for Poincare GloVe is more efficient than HNSW over Euclidean embeddings, this is but 1 dataset and 1 graph construction. To make a worthwhile empirical claim, it is necessary to provide results for at least a few datasets and a few graph constructions.

6. For all empirical results above, I did not see any mention of trials/mean/standard deviation. It is my understanding that graph-based NNS starts with some random location, and then propagates through the graph, guided by the metric. Hence, results may vary considerably from trial to trial. What was done in the paper? Were several trials run? What were the means/standard deviations? Please either report this information or re-run experiments for several trials and then report said information.

7. Although I have alluded to this above already, I would like to reiterate that given that the method is generally applicable to various graph constructions, I would have liked to see at least some results comparing the performance of graph-based NNS with graph constructions other than HNSW. Although HNSW is considered to be one of the best similarity graphs for NNS (as you claim), my understanding is that this was the case for Euclidean (i.e. non-hyperbolic) NNS. Given that the paper now considers the setting of hyperbolic graph-based NNS, a different graph construction may perhaps be better. Regardless, seeing a comparison of several (e.g. at least 3) graph constructions in an empirical setting would considerably help with ascertaining HNSW as a good graph construction for this setting.

8. I would like to see more than a toy experiment (given in Section 5.3) to ascertain the results for the dense setting. The application of the dense theoretical results as is seems incomplete to me. Perhaps some analog of what was done in Prokhorenkova & Shekhovtsov (2020) is possible to obtain a real application of the theoretical dense results?

### Additional Comments

The paper writing was, for the most part, adequate. A non-exhaustive list of minor corrections/suggestions is given below:

- "opposite approach" in the introduction is not really well-defined. "alternative approach" would be more appropriate.
- For the first "$nM^d$" mentioned on page 3, it is unclear what $M$ is (i.e. whether it has some geometric meaning). If it is just a constant, simply say "where M is some constant". Moreover, I believe you overload the usage of $M$ to be both a Riemannian manifold and a constant. I would recommend using a calligraphic $\mathcal{M}$ to represent a Riemannian manifold.
- The following claim is made several times: "...hyperbolic geometry was recently found to be more suitable for data representation in various domains...". I would caution this by saying that hyperbolic geometry was shown to be more suitable for hierarchical data (Nickel & Kiela 2017, 2018), not necessarily data representation in general. Please introduce the necessary qualifiers when making such claims.

**Summary Of The Paper:**

The paper considers the setting of nearest neighbor search (NNS) and devises a graph-based algorithm for this task in Hyperbolic space. The problem is pertinent, as hyperbolic space has become prevalent for representing hierarchical data in various domains, and nearest neighbor search is a very fundamental problem. The paper proposes a graph-based nearest neighbor search closely based on existing literature, and analyzes time and space complexity assuming a uniformly distributed dataset of $n$ elements over a $d$-dimensional ball of radius $R$; the dataset is assumed to be dense in the underlying space ($\log n \gg d$). It is shown that under some assumptions on $d$ and $R$, graph-based NNS has lower time complexity in hyperbolic space when compared with its Euclidean counterpart. Practically, the paper focuses on the sparse data setting. It is shown that graph-based NNS is more efficient for Poincaré GloVe word embeddings than for Euclidean GloVe word embeddings, assuming an HNSW graph construction. Moreover, it is shown that the proposed graph-based NNS method outperforms other existing baselines on Poincaré GloVe embeddings.

**Summary Of The Review:**

### Verdict

This paper developed an extension of the Euclidean graph-based NNS from Prokhorenkova & Shekhovtsov (2020) to hyperbolic space. Although the problem is relevant, the paper as is lacks considerably, both in terms of theoretical and empirical results. Namely, it does not address the sparse case theoretically, and the theory that is developed borrows heavily from Prokhorenkova & Shekhovtsov (2020); the necessary analytic constructions/proof techniques necessary to bridge the gap are fairly straightforward, and cast considerable shade on claims of theoretical novelty. Moreover, empirical results lack considerably. First, non-toy results were given only for the sparse setting, not the dense setting (which was the only setting for which theory was provided). The uniformity assumption and assumption of small dimension was not addressed in the application section. These assumptions were critical for a proper theoretical analysis, and should in some way be handled when the theory is applied. Only a single baseline was given for Section 5.1, and claims of hyperbolic graph-based NNS efficiency (over the Euclidean counterpart) were made by testing on 1 dataset with 1 graph construction. A considerably more thorough evaluation is required. Trials/mean/standard deviation were not reported, despite this being crucial for graph-based NNS. Only a single graph construction was considered throughout the entirety of the empirical evaluation, despite the fact that the tested setting (hyperbolic graph-based NNS) does not have a previously established state-of-the-art graph construction. All of this taken into account, I must recommend a reject rating for this paper.

### Updated Rating

In their revised paper, the authors have assuage some of my concerns as they have been laid out above. As a result, I have updated my rating to a 5/10. Given that the changes were fairly considerable, I hesitate to recommend acceptance until they have been carefully evaluated; moreover, I believe more can be done to address the concerns reviewers have, specifically surrounding empirical results.

---

> ### Author Response · Authors · 2021-11-12
> **Reply part 1: our plans for additional experiments**
>
> We would like to thank the reviewer for the thorough review. We are ready to cooperate in improving the paper and will do our best during the discussion period.
>
> Regarding the comments 5 and 7, we agree that additional experiments with other NNS graph-based methods and on other datasets may stronger support the claim that “Graph-based algorithms are well suited for NNS in hyperbolic spaces in practice” (the goal of Sections 5.1-5.2). Therefore, during the discussion period, we plan to perform the following experiments:
> - Add more graph-based algorithms. We are planning to add NSG (Fu et al., 2019) and kNN with beam search and long edges from Prokhorenkova & Shekhovtsov, 2020. The aim here is to show that all the algorithms work reasonably well.
> - Add more datasets. During the discussion period, we plan to add a dataset for another domain (recommendations) to see whether our conclusions generalize.
> - We also want to consider GloVe and Poincare GloVe embeddings for smaller dimensions (e.g., d=10). The aim here is to obtain experimental results in a dense setting that better corresponds to our theoretical analysis.
>
> We need some time to carefully perform these experiments. Thus, we thought it would be appropriate to ask whether such additional empirical evaluation can be convincing and may change your opinion regarding the value of the work (assuming that the obtained conclusions agree with the expectations).
>
> Detailed comments regarding the remaining concerns are given in the subsequent comments.
>
> Also, we thank the reviewer for the additional comments, which will be taken into account in the revised version.

---

> ### Author Response · Authors · 2021-11-12
> **Reply part 2: addressing concerns regarding theoretical analysis**
>
> **1.** We agree that it would be nice to cover all possible scenarios. However, the sparse uniform setting seems to be both theoretically degenerate and practically less realistic. First, let us explain why we think that the dense setting is realistic and practically interesting:
> - Several papers argue that hyperbolic spaces, as well as some more complex constructions, are beneficial when the dimension is small [1,2]. For larger dimensions, improvements obtained with hyperbolic embeddings are often less impressive. Thus, the low-dimensional scenario is natural for hyperbolic embeddings and thus should be the first setting to consider.
> - Even when the dataset is embedded in a high-dimensional space, it usually has a much smaller local intrinsic dimension. Informally speaking, a dataset can be locally approximated by a close-to-uniform distribution in a lower-dimensional space, and thus our theoretical results can be readily applied.
> - In fact, Prokhorenkova & Shekhovtsov (2020) mainly focus on the dense regime - their results on beam search and long edges are proven for this regime. In Appendix F.1, the authors provide additional arguments on the dense regime being reasonable.
> - Also, according to Prokhorenkova & Shekhovtsov (2020), in the sparse regime, long edges are not helpful since graph-based NNS is expected to converge in just two steps (for the uniform distribution). This contradicts numerous practical observations, thus implying that the dense setup is more realistic.
> - Similarly, the optimal graph degree observed in practice better agrees with the theoretical guarantees in the dense regime.
>
> [1] Gu A. et al. Learning mixed-curvature representations in product spaces. ICLR, 2018.
>
> [2] Shevkunov K., Prokhorenkova L. Overlapping Spaces for Compact Graph Representations. NeurIPS 2021.
>
> Regarding theoretical analysis for the sparse regime, we partially address this comment in our reply to Reviewer 6ftD. When we have $d \gg \log n$, all elements are located within a narrow spherical shell of a width of order $\frac{\log n}{d} \ll 1$ near the boundary of the ball. Thus, one may suggest that the behavior of NNS will be similar to that on the sphere, making this situation degenerate. On the other hand, when $d \gg \log n$, all pairwise angles between the elements are very close to $\pi/2$. We plan to work on this during the discussion period. We also plan to add an experiment on synthetic data in the sparse regime to illustrate this setting.
>
> **2.** Regarding the proofs, we deliberately use the notation of Prokhorenkova & Shekhovtsov (2020) to show similarities and differences between Euclidean and hyperbolic settings. In our opinion, the most interesting part of the analysis is figuring out in which parameter regimes and under which conditions NNS in hyperbolic spaces can provably work well (and why this happens). We did our best to describe our intuition in Section 4. When this is figured out, the remaining proof is technical. However, we still had to perform this technical proof to get formally correct results. Therefore, we place these details in the Appendix. In summary, while some of the technical details are similar to those in Prokhorenkova & Shekhovtsov (2020), we respectfully disagree with the claim that “A large portion of the paper lacks novelty” - the considered task is novel, there is an additional parameter to consider, and the obtained complexity is different compared to the Euclidean case.

---

> ### Author Response · Authors · 2021-11-12
> **Reply part 3: addressing remaining concerns regarding experiments**
>
> **3.** First, let us stress that our experiments in Section 5.3 are performed in the setting corresponding to our theoretical analysis. Section 5.2 aimed to show that graph-based approaches can be applied not only in cases covered by our theoretical results. While the applicability of graph-based NNS is widely acknowledged for the Euclidean space (due to extensive empirical analysis), this is not known for hyperbolic spaces. Additional experiments that we plan to conduct (described in the previous comment) will help us to illustrate this better.
>
> Regarding connection with theory, it is suggested in the literature that real datasets have relatively low intrinsic dimensions. Thus, datasets can be considered locally dense, while the density may vary within one dataset. This is very hard to formalize rigorously; however, intuitively, our theoretical results can be applied since convergence depends on local properties. We will think about additional measurements that may show that the considered datasets are indeed locally dense in most regions and thus partially bridge the gap between theory and practice.
>
> **8.** As we pointed out above, our aim in Section 5.2 is to show that graph-based approaches can be applied not only in cases covered by our theoretical results but also in practice. We agree that to illustrate this better, additional experiments may be performed, but we do not see how dimensionality reduction + uniformization would help here. As Prokhorenkova & Shekhovtsov (2020) show, for the Euclidean space, this technique improves simple kNN graphs, but HNSW still has better performance. Thus, while dimensionality reduction seems to be promising, it is not the choice for practitioners for now. What we can do instead is to consider Poincare GloVe embeddings for a lower dimension (instead of 100 considered in the paper). Experiments with this dataset will better correspond to the dense setting analyzed theoretically.
>
> **4.** We fully agree that currently, the field of hyperbolic NNS lacks strong baselines. However, we are not sure if including Algorithms 1 and 2 from Wu & Charikar (2020) will improve the soundness of our empirical evaluation. If required, we will do our best to add them, but this experiment seems to have a lower priority than the other experiments we plan. On theoretical grounds, we expect the performance of the remaining baselines to be considerably worse. To explain our intuition, let us note that both Algorithms 1 and 2 require the exact Euclidean NNS base algorithm, which is infeasible for realistic settings. However, assume that instead of the exact algorithm, we consider a good approximate one (with high recall). We can, for example, take Euclidean HNSW that is known to be good for Euclidean embeddings. In this case, Algorithm 1 suggests applying Euclidean HNSW (to points within the Poincare ball) several times, while the query point is moved to get better hyperbolic neighbors for each next run. Since at least two runs of the Euclidean HNSW are needed, we expect the complexity to be at least two times larger. Moreover, there is no need to apply Euclidean HNSW to Poincare points when we can directly apply its hyperbolic version and use only one run of the algorithm. This is why we expect such baselines to be at least twice slower compared to a simple solution of applying graph-based approaches directly.
>
> **6.** Our current results show the average number of distance computations per query. Here we need to stress that not all algorithms use random starts. In particular, HNSW has a particular fixed starting node. Thus, we cannot measure standard deviation wrt random starts for HNSW. On the other hand, Spherical Shell and the algorithms in Section 5.3 are randomized. In our experiments, we have not observed any noticeable difference between different runs of these algorithms. We can add the corresponding numbers, but we cannot do this for HNSW, as written above.

---

> ### Author Response · Authors · 2021-11-22
> **Revised paper**
>
> We’ve updated the paper.
> * Addressing your concerns regarding the theory, we added the analysis of the sparse regime. In this case, we show that the elements are very close to the border of the ball, so complexity can formally be transferred from the spherical space. This additionally indicates that the dense setting seems to be more interesting.
> * We added more datasets. Namely, we now have GloVe embeddings for d=10 and an additional dataset for the recommendation problem.
> * We added the NSG graph-based algorithm. NSG and HNSW are known to have comparable performance for the Euclidean space, while HNSW often has better performance for the hyperbolic case. Both of them outperform the baseline.
>
> Thank you again for the thorough review of our paper! We look forward to further discussions.

---

> > ### Comment · Reviewer_1ri4 · 2021-11-29
> > **Updated Rating**
> >
> > I would like to thank the authors for putting in considerable effort to assuage my concerns as they have been laid out above. They have been partially assuaged. As a result, I have updated my rating to a 5/10. Given that the changes were fairly considerable, I hesitate to recommend acceptance until they have been carefully evaluated; moreover, I believe more can be done to address the concerns reviewers have, specifically surrounding empirical results.

---

### Author Response · Authors · 2021-11-23
**Rebuttal Revision Summary**

We would like to thank the reviewers for their work! We modified the paper according to the comments and suggestions.

To sum up:
* We added the theoretical analysis of the sparse regime. So, now we have both dense and sparse regimes analyzed. We agree that this is important for the completeness of the study. The new results can be found in Section 4.3 and Appendix E.
* We also significantly extended the experimental part. In particular, we added one more graph-based method, so now we have HNSW and NSG, both are known to be state-of-the-art for the Euclidean space. We also have two additional datasets: Poincare GloVe with a smaller dimension (d=10) and the dataset for a recommendation problem that has different nature and properties. The new results agree with our claims and can be found in Section 5.1 and Appendix F.
* We made several improvements throughout the text according to the comments of the reviewers. In particular, we added more motivation and discussion on the relation between our theoretical assumptions and practice.

To sum up, our paper provides theoretical and practical explanations for how and why graph-based methods work in hyperbolic spaces. We believe that our research makes a valuable contribution to the community and hope that it will attract more attention to the analysis of NNS in hyperbolic spaces and, in particular, to graph-based methods. Graph-based methods are appealing since they are universal, easy to use, and are efficiently implemented in several libraries.

We are looking forward to further discussions! We will be happy to answer any questions and provide additional details.

---

### Decision · Program_Chairs · 2022-01-20

**Decision:**

Accept (Poster)

**Comment:**

The paper studies the nearest-neighbor search problem for objects embedded in hyperbolic space. It develops a graph-based approach to NNS in hyperbolic space, showing (interestingly) that the time complexity of graph-based NNS can be lower in hyperbolic space than in Euclidean space under some assumptions. This nice theoretical analysis feels like an intuitive result, as hyperbolic space is in some sense more graph-like/tree-like then Euclidean space, so it is not too surprising for graph-based methods to perform better here. What's really cool about this theoretical result is to see something that is _easier_ in hyperbolic space—normally things are harder in hyperbolic space due to its great volumes and significant curvature. The discussion among reviewers was mostly positive, with the majority of reviewers leaning towards acceptance and multiple reviewers expressing that the author response and revision addressed their concerns and raising their scores accordingly. There is some question as to whether the revision is too large and should go through peer review again, but on balance I think it is close enough to the original submission that it can be reasonably accepted here. Therefore, following the majority opinion of the reviewers, I will recommend acceptance.